# New versions of maps and connected spaces via supra soft sd-operators

**Alaa M. Abd El-latif**[1], **A. A. Azzam**[2,3]*, **Radwan Abu-Gdairi**[4], **M. Aldawood**[2], **Mesfer H. Alqahtani**[5]

**1** Mathematics Department, College of Sciences and Arts, Northern Border University, Rafha, Saudi Arabia, **2** Department of Mathematics, Faculty of Science and Humanities, Prince Sattam Bin Abdulaziz University, Alkharj, Saudi Arabia, **3** Department of Mathematics, Faculty of Science, New Valley University, Elkharga, Egypt, **4** Mathematics Department, Faculty of Science, Zarqa University, Zarqa, Jordan, **5** Department of Mathematics, University College of Umluj, University of Tabuk, Tabuk, Saudi Arabia

\* azzam0911@yahoo.com

**Data Availability Statement:** All relevant data are within the paper.

**Funding:** The authors extend their appreciation to the Deanship of Scientific Research at Northern Border University, Arar, KSA for funding this

## Abstract

In this manuscript we use novel types of soft operators to define new approaches of soft maps in the frame of supra soft topologies (or SSTSs), namely supra soft somewhere dens continuous (or SS-sd-continuous), SS-sd-open and SS-sd-closed maps. With the help of SS-closure (interior) operators and SS-sd-closure (interior) operators we succeed to introduce many equivalent conditions and several important properties to these notions. To name a few: We prove that there is an one to one between the SS-sd-open and SS-sd-closed maps under a bijective soft map, supported by counterexample to confirm the necessity of the bijectivity condition. Furthermore, we present the concept of SS-sd-separated sets with intersected characterizations, as a prelude to studying the connectedness in a supra soft topological space (or SSTS). Moreover, we show that, there is no priori relationship between supra soft-sd-connectedness in an SSTS and its parametric supra topological spaces in general, supported by concrete counterexamples. Finally, we prove that the image of an SS-sd-connected set under an SS-sd-irresolute map is an SS-sd-connected.

## 1 Introduction

Mashhour et al. [1] presented the definition of supra topological spaces by ignoring the condition of finite intersections in the classical topologies definition. Several applications to this research have been introduced in [2–4]. Certain rough sets models based on supra topological spaces have been defined in [5]. Al-Odhari [6], in 2015, defined the infra-topological spaces concept, which investigated in [7–9].

The soft set theory [10] plays an important alternative tool of rough, fuzzy and crisp theories whose they have some problems with uncertainties. Maji et al. [11] improved and investigated it by introducing more operations. Several concrete applications were introduced in decision making problem [12], rough set models [13, 14] and medical sciences [15]. The maps have important roles in topological spaces, not only as a tool to determine which topological properties are preserved, but also to study the classification of spaces by maps and reversely.

research work through the project number "NBU-FFR-2024-2727-03". This study is also supported via funding from Prince Sattam bin Abdulaziz University project number (PSAU/2024/R/1445) and this research is funded partially by Zarqa University Jordan.

**Competing interests:** The authors have declared that no competing interests exist.

The concepts of soft continuity [16] have been introduced in 2001, which extended in [17–19]. The notion of soft topological space (or STS) [20, 21] were first introduced in 2011. After that many scientists explored many types of weaker soft open sets like, soft pre- (respectively, $\beta$-, $\alpha$-) open sets [22, 23], soft semi-open sets [24–26], soft b-open sets [27, 28], nearly soft $\beta$-open sets [29], soft somewhat open sets [30], soft sd-sets [31] and finite soft-open sets [32]. More investigation to the soft somewhere dense continuity were introduced in [33]. Azzam et al. [34] used soft set operators to generate new soft topologies. An application on the concept of soft somewhat open sets in compactness and connectedness was introduced in [35].

In 2014, the approaches of SSTSs [36] have been introduced. Also, they presented different types of SS-continuous maps. Moreover the relationships among them have been studied. Later, the definition of SS-b-continuous (respectively, open, closed) maps [37] were introduced in 2015. Recently, Abd El-latif [38] presented the concepts of SS-$\delta_i$-continuity in 2024. Moreover, he presented the notions of SS-sd-sets, SS-sd-closure (respectively, interior, cluster) operators [39]. After that, he and his co-authors [40] used the SS-sd-closure operator to generalize several famous notions. New decomposition of supra soft locally closed sets and supra slc-continuity have been introduced in [41, 42]. Separation axioms in the frame of SSTSs were studied in [43, 44]. Connectedness [45] has an important role in discriminating between different soft topological spaces, which investigated in [46–48]. Hussain [49] applied it to decision making problems. Kandil et al. [50] generalized this notion by using the soft ideal approach [51–53]. In 2020, the connectedness [54] via soft sd-sets have been presented. Al-Ghour and Al-Saadi [55] presented the notion of soft weakly connected sets in 2023. Abd El-latif [56, 57] introduced the connectedness to SSTS. The notions of supra soft topological ordered spaces have been introduced in [58, 59].

This project is devoted to define novel approaches of soft continuity in SSTSs inspired by soft point and SS-sd-sets, named SS-sd-continuous maps, in section 3. Moreover, we define and study new interesting properties of the SS-sd-open (closed) maps. In especial, we prove that, there is an one to one between the SS-sd-open and SS-sd-closed maps under a bijective soft map, supported by counterexample to confirm the necessity of the bijectivity condition. Furthermore, we provide many equivalent conditions to these concepts with the help of SS-closure operator, SS-interior operator, SS-codense sets and SS-nowhere dense sets. In section 4, we introduce the concept of SS-sd-separated sets as an extension to their corresponding in [56] Also, we discuss many of its characteristics such as, we show that the pre-image of SS-sd-separated sets under a surjective SS-sd-irresolute map is an SS-sd-separated. In addition, we use it to introduce the connectedness in SSTS. We discuss its essential properties in detail. Finally, many concrete examples and counterexamples are provided.

## 2 Preliminaries

In this section, we introduce the notions and terminologies which will needed in this manuscript, for more details see [16, 20, 36, 39].

**Definition 2.1** [10] *Let $\Delta$ is a set of parameters and $U$ is the universe set. A pair $(T, \Delta)$; denoted by $T_\Delta$, is called a soft set, where $T_\Delta = \{T(\gamma): \gamma \in \Delta, T: \Delta \to P(U)\}$. If $T(\gamma) = U$ $(T(\gamma) = \varphi)$ for all $\gamma \in \Delta$, then $(T, \Delta)$ is called an absolute (a null) soft set and will denoted by $\tilde{U}$ $(\tilde{\varphi})$. Henceforth, the family of all soft sets will denoted by $S(U)_\Delta$.*

**Definition 2.2** [20] *The class $\tau \subseteq S(U)_\Delta$ is called an STS on $U$ if $\tau$ contains $\tilde{U}, \tilde{\varphi}$ and closed under finite soft intersection and arbitrary soft union. Also, each $(G, \Delta) \in \tau$ is called soft open set, and its soft complements is called soft closed.*

**Definition 2.3** [20] *Let $(U, \tau, \Delta)$ be an STS and $(T, \Delta) \in S(U)_\Delta$. The soft closure (interior) of $(T, \Delta)$; is denoted by $cl(T, \Delta)$ $(int(G, \Delta))$, is the soft intersection (union) of all soft closed supersets (open subsets) of $(T, \Delta)$.*

**Definition 2.4** [20, 24] *Let $(G, \Delta) \in S(U)_\Delta$. If there are $s \in U$ and $\gamma \in \Delta$ such that $G(\gamma) = \{s\}$ and $G(\gamma') = \varphi$ for each $\gamma' \in \Delta - \{\gamma\}$, then $(G, \Delta)$ is called a soft point in $\tilde{U}$, and will denoted by $s_\gamma$. Also, $s_\gamma \tilde{\in} (F, \Delta)$, if for $\gamma \in \Delta$ we have that $G(\gamma) \subseteq F(\gamma)$.*

**Theorem 2.5** [16] *The following statements satisfy for the soft map $\psi_{sd}$: $(U, \tau, \Delta) \to (V, \sigma, \Lambda)$:*

(1) $\psi_{sd}^{-1}((N^{\tilde{c}}, \Lambda)) = (\psi_{sd}^{-1}(N, \Lambda))^{\tilde{c}} \; \forall \; (N, \Lambda) \in S(V)_\Lambda$.

(2) $\psi_{sd}(\psi_{sd}^{-1}((N, \Lambda))) \tilde{\subseteq} (N, \Lambda) \; \forall \; (N, \Lambda) \in S(V)_\Lambda$. *The equality holds if $\psi_{sd}$ is surjective.*

(3) $(M, \Delta) \tilde{\subseteq} \psi_{sd}^{-1}(\psi_{sd}((M, \Delta))) \; \forall \; (M, \Delta) \in S(U)_\Delta$. *The equality holds if $\psi_{sd}$ is injective.*

(4) $\psi_{sd}(\tilde{U}) \tilde{\subseteq} \tilde{V}$. *The equality holds if $\psi_{sd}$ is surjective.*

**Definition 2.6** [36] *The class $\mu \subseteq S(U)_\Delta$ is called an SSTS on U if $\mu$ closed under arbitrary soft union and contains $\tilde{U}, \tilde{\varphi}$. Also, each $(G, \Delta) \in \mu$ is called SS-open set and its soft complements is called SS-closed.*

*Also, the SS-interior (closure) of a soft subset $(G, \Delta)$; is denoted by $int^s(G, \Delta)$ $(cl^s(G, \Delta))$, is the soft union (intersection) of all SS-open subsets (closed supersets) of $(G, \Delta)$.*

**Definition 2.7** [36] *Let $((U, \mu, \Delta)$ be an SSTS, then $(G, \Delta) \in S(U)_\Delta$ is called SS-semi open set if $(G, \Delta) \tilde{\subseteq} cl^s(int^s(G, \Delta))$. Also, $(G^{\tilde{c}}, \Delta)$ is called SS-semi-closed.*

**Definition 2.8** [36] *If $\tau \subset \mu$ for an SSTS $(U, \mu, \Delta)$ and STS $(U, \tau, \Delta)$, then $\mu$ is called an associated SSTS with $\tau$.*

**Definition 2.9** [36] *A soft map $\psi_{sd}$: $(U, \tau, \Delta) \to (V, \sigma, \Lambda)$ with $\tau \subset \mu$ is called SS-continuous if $\psi_{sd}^{-1}(T, \Lambda) \in \mu \; \forall \; (T, \Lambda) \in \sigma$.*

**Definition 2.10** [39] *A soft subset $(T, \Delta)$ of an SSTS $(U, \mu, \Delta)$ is called SS-sd-set if $int^s(cl^s(T, \Delta)) \neq \tilde{\varphi}$. Also, $(T^{\tilde{c}}, \Delta)$ is called SS-sc-set. We denote the class of all SS-sd-sets (SS-sc-sets) by $SD(U)_\Delta$ $(SC(U)_\Delta)$. Also, if $int^s(cl^s(T, \Delta)) = \tilde{\varphi}$, then it is called SS-nowhere dense.*

**Proposition 2.11** [39] *A soft subset $(T, \Delta)$ of an SSTS $(U, \mu, \Delta)$ is either SS-sd-set or SS-sc-set.*

**Definition 2.12** [39] *The SS-sd-closure (interior) for a soft subset $(T, \Delta)$ of an SSTS $(U, \mu, \Delta)$; is denoted by $cl^s_{sd}(T, \Delta)$ $(int^s_{sd}(T, \Delta))$, is the smallest (largest) SS-sc-supersets (SS-sd-subsets) of $(T, \Delta)$.*

**Theorem 2.13** [39] *Let $(U, \mu, \Delta)$ be an SSTS and $(T, \Delta) \in S(U)_\Delta$, we have that*

(1) $cl^s_{sd}(T^{\tilde{c}}, \Delta) = [int^s_{sd}(T, \Delta)]^{\tilde{c}}$ *and* $int^s_{sd}(T^{\tilde{c}}, \Delta) = [cl^s_{sd}(T, \Delta)]^{\tilde{c}}$.

(2) $cl^s_{sd}(T, \Delta) \tilde{\subseteq} cl^s(T, \Delta)$.

(3) $int^s(T, \Delta) \tilde{\subseteq} int^s_{sd}(T, \Delta)$.

## 3 Novel categories of soft maps via supra soft sd-sets

In this section, we introduce new approaches to soft continuity in SSTSs inspired by soft point and supra soft sd-sets, named SS-sd-cts. Furthermore, more characterizations to the SS-sd-open (closed) maps have been studied. Specifically, we prove that, there is an one to one between the SS-sd-open and SS-sd-closed maps under a bijective soft map, supported by counterexample to confirm the necessity of the bijectivity condition. With the help of SS-closure operator, SS-interior operator, SS-codense sets and SS-nowhere dense sets we provid many equivalent conditions to these concepts.

**Definition 3.1** *A soft map $\psi_{sd}$: $(U, \tau, \Delta) \to (V, \sigma, \Lambda)$ with $\tau \subset \mu$ is said to be SS-sd-cts at the soft point $s_\gamma \tilde{\in} \tilde{U}$ if for each $(G, \Lambda) \in \sigma$ containing $\psi_{sd}(s_\gamma)$, there exists $(H, \Delta) \in SD(U)_\Delta$ containing $s_\gamma$ such that*

$$\psi_{sd}(H, \Delta) \tilde{\subseteq} (G, \Lambda).$$

*If $\psi_{sd}$ is SS-sd-cts for each $s_\gamma \tilde{\in} \tilde{U}$, then it is said to be SS-sd-cts.*

**Theorem 3.2** *A soft map $\psi_{sd}$: $(U, \tau, \Delta) \to (V, \sigma, \Lambda)$ with $\tau \subset \mu$ is SS-sd-cts iff either $\psi_{sd}^{-1}(G, \Lambda) = \tilde{\varphi}$ or $\psi_{sd}^{-1}(G, \Lambda) \in SD(U)_\Delta$ for each $(G, \Lambda) \in \sigma$.*

**Proof**. "$\Rightarrow$" Let $(K, \Lambda) \in \sigma$. If $\psi_{sd}^{-1}(K, \Lambda) = \tilde{\varphi}$, then we get the result. If $\psi_{sd}^{-1}(K, \Lambda) \neq \tilde{\varphi}$. It follows that,

for every $\psi_{sd}(s_\gamma) \tilde{\in} (K, \Lambda)$, there exists $(H, \Delta) \in SD(U)_\Delta$ containing $s_\gamma$ such that $\psi_{sd}(H, \Delta) \tilde{\subseteq} (G, \Lambda)$, from the hypothesis,

which follows that,

$$s_\gamma \tilde{\in} (H, \Delta) \tilde{\subseteq} \psi_{sd}^{-1}(K, \Lambda).$$

Hence,

$$\psi_{sd}^{-1}(K, \Lambda) = \tilde{\bigcup}\{(H, \Delta): \ s_\gamma \tilde{\in} (H, \Delta) \tilde{\subseteq} \psi_{sd}^{-1}(K, \Lambda)\}.$$

Thus, $\psi_{sd}^{-1}(K, \Lambda) \in SD(U)_\Delta$.

"$\Leftarrow$" Suppose that $\psi_{sd}(s_\gamma) \tilde{\in} (K, \Lambda) \in \sigma$, for any soft point $s_\gamma \tilde{\in} \tilde{U}$, then

$$s_\gamma \tilde{\in} \psi_{sd}^{-1}(K, \Lambda) \neq \tilde{U}, \psi_{sd}^{-1}(K, \Lambda) \in SD(U)_\Delta.$$

Hence,

$\psi_{sd}(\psi_{sd}^{-1}(K, \Lambda)) \tilde{\subseteq} (K, \Lambda)$, and so $\psi_{sd}$ is SS-sd-cts at $s_\gamma \tilde{\in} \tilde{U}$.

Therefore, $\psi_{sd}$ is an SS-sd-cts, from Definition 3.1.

**Corollary 3.3** *A bijective soft map $\psi_{sd}$: $(U, \tau, \Delta) \to (V, \sigma, \Lambda)$ with $\tau \subset \mu$ is SS-sd-cts iff $\psi_{sd}^{-1}(G, \Lambda) \in SD(U)_\Delta$ for each $(G, \Lambda) \in \sigma$.*

**Proof**. Follows from Theorem 3.2.

**Corollary 3.4** [39] *Let $(U, \mu, \Lambda)$ be an SSTS and $(K, \Delta) \in S(U)_\Delta$. Then, $(K, \Delta) \in SD(U)_\Delta$ iff there exists $\tilde{\varphi} \neq (O, \Delta) \in \mu$ such that $(O, \Delta) \tilde{\subseteq} cl^s(K, \Delta)$. Also, $(K, \Delta) \in SC(U)_\Delta$ iff there exists a proper SS-closed set $(H, \Delta)$ such that $int^s(K, \Delta) \tilde{\subseteq} (H, \Delta)$.*

**Theorem 3.5** *The following are equivalent for soft map $\psi_{sd}$: $(U, \tau, \Delta) \to (V, \sigma, \Lambda)$ with $\tau \subset \mu$:*

(1) *$\psi_{sd}$ is SS-sd-cts;*

(2) *There exists non-null SS-open subset $(G, \Delta)$ of $\tilde{U}$ such that $(G, \Delta) \tilde{\subseteq} cl^s[\psi_{sd}^{-1}(E, \Lambda)]$, for each $(E, \Lambda) \in \sigma$ in which $\psi_{sd}^{-1}(E, \Lambda) \neq \tilde{\varphi}$;*

(3) *There exists a proper SS-closed subset $(G, \Delta)$ of $\tilde{U}$ such that $int^s[\psi_{sd}^{-1}(E, \Lambda)] \tilde{\subseteq} (G, \Delta)$, for each $(E, \Lambda) \in \sigma^c$ in which $\psi_{sd}^{-1}(E, \Lambda) \neq \tilde{U}$;*

(4) *$\psi_{sd}(G, \Delta)$ is SS-dense subset of $\psi_{sd}(\tilde{U})$, for each $(G, \Delta)$ is SS-open and SS-dense subset of $\tilde{U}$.*

**Proof**.

(1) $\Rightarrow$ (2) Let $(E, \Lambda) \in \sigma$ such that $\psi_{sd}^{-1}(E, \Lambda) \neq \tilde{\varphi}$, then $\psi_{sd}^{-1}(E, \Lambda) \in SD(U)_\Delta$ from Theorem 3.2. It follows that, there exists $\tilde{\varphi} \neq (G, \Delta) \in \mu$ such that $(G, \Delta) \tilde{\subseteq} cl^s[\psi_{sd}^{-1}(E, \Lambda)]$, from Corollary 3.4.

(2) $\Rightarrow$ (3) Let $(E, \Lambda) \in \sigma^c$ such that $\psi_{sd}^{-1}(E, \Lambda) \neq \tilde{U}$, then $(E^{\tilde{c}}, \Lambda) \in \sigma$ with $\psi_{sd}^{-1}(E^{\tilde{c}}, \Lambda) \neq \tilde{\varphi}$. It follows that, there exists $\tilde{\varphi} \neq (G, \Delta) \in \mu$ such that

$$(G, \Delta) \tilde{\subseteq} cl^s(\psi_{sd}^{-1}(E^{\tilde{c}}, \Lambda)) = [int^s(\psi_{sd}^{-1}(E, \Lambda))]^{\tilde{c}}, \text{ from (2),}$$

which follows $int^s(\psi_{sd}^{-1}(E, \Lambda)) \tilde{\subseteq} (G^{\tilde{c}}, \Delta)$, $(G^{\tilde{c}}, \Delta)$ is proper SS-closed subset of $\tilde{U}$. Thus, the needed result is obtained.

(3) $\Rightarrow$ (4) Let $(G, \Delta)$ be both SS-dense and SS-open subset of $\tilde{U}$. Suppose conversely, $\psi_{sd}(G, \Delta)$ is not SS-dense subset of $\psi_{sd}(\tilde{U})$. That is,

$$cl^s(\psi_{sd}(G, \Delta)) \neq \tilde{V}.$$

It follows that, there exists a proper SS-closed subset $(S, \Lambda)$ of $\tilde{V}$ such that $\psi_{sd}(G, \Delta) \tilde{\subseteq} (S, \Lambda)$, and so $(G, \Delta) \tilde{\subseteq} \psi_{sd}^{-1}(S, \Lambda)$. Since $(G, \Delta)$ is SS-open,

$$(G, \Delta) \tilde{\subseteq} int^s(\psi_{sd}^{-1}(S, \Lambda)) \tag{1}$$

By (3), there exists a proper SS-closed subset $(H, \Delta)$ of $\tilde{U}$ such that

$$int^s[\psi_{sd}^{-1}(S, \Lambda)] \tilde{\subseteq} (H, \Delta) \tag{2}$$

From Eqs (1) and (2), $(G, \Delta) \tilde{\subseteq} int^s[\psi_{sd}^{-1}(S, \Lambda)] \tilde{\subseteq} (H, \Delta)$, $(H, \Delta) \neq \tilde{U}$, which contradicts that $(G, \Delta)$ is SS-dense. Therefore, $\psi_{sd}(G, \Delta)$ is SS-dense subset of $\psi_{sd}(\tilde{U})$.

(4) $\Rightarrow$ (1) Let $(Z, \Lambda) \in \sigma$. If $\psi_{sd}^{-1}(Z, \Lambda) = \tilde{\varphi}$, then we get (1). If $\psi_{sd}^{-1}(Z, \Lambda) \neq \tilde{\varphi}$. Suppose conversely, $\psi_{sd}^{-1}(Z, \Lambda) \notin SD(U)_\Delta$, which follows

$$int^s(cl^s(\psi_{sd}^{-1}(Z, \Lambda))) = \tilde{\varphi},$$

and so

$$cl^s(int^s(\psi_{sd}^{-1}(Z^{\tilde{c}}, \Lambda))) = [int^s(cl^s(\psi_{sd}^{-1}(Z, \Lambda)))]^{\tilde{c}} = \tilde{U}.$$

Hence, $int^s(\psi_{sd}^{-1}(Z^{\tilde{c}}, \Lambda))$ is both SS-open and SS-dense subset of $\tilde{U}$. From (4), $\psi_{sd}[int^s(\psi_{sd}^{-1}(Z^{\tilde{c}}, \Lambda))]$ is SS-dense subset of $\psi_{sd}(\tilde{U})$.

Therefore,

$$cl^s(\psi_{sd}[int^s(\psi_{sd}^{-1}(Z^{\tilde{c}}, \Lambda))]) = \psi_{sd}(\tilde{U}).$$

Thus,

$\psi_{sd}(\tilde{U}) = cl^s(\psi_{sd}[int^s(\psi_{sd}^{-1}(Z^{\tilde{c}}, \Lambda))]) \tilde{\subseteq} cl^s(\psi_{sd}[\psi_{sd}^{-1}(Z^{\tilde{c}}, \Lambda)]) \tilde{\subseteq} cl^s(Z^{\tilde{c}}, \Lambda) = (Z^{\tilde{c}}, \Lambda)$, and so $(Z, \Lambda) = \tilde{\varphi}$,

which contradicts our assumption, and follows $\psi_{sd}^{-1}(Z, \Lambda) \in SD(U)_\Delta$, then we get (1).

**Definition 3.6** *A soft subset $(K, \Delta)$ of an SSTS $(U, \mu, \Delta)$ is called SS-dense set if* $cl^s(K, \Delta) = \tilde{U}$. *Also, it is called SS-codense set if $int^s(K, \Delta) = \tilde{\varphi}$.*

**Theorem 3.7** *Let $\psi_{sd}$: $(U, \tau, \Delta) \to (V, \sigma, \Lambda)$ be a bijection soft map with $\tau \subset \mu$ and $\sigma \subset \mu^*$, then the following are equivalent:*

(1) $\psi_{sd}$ is SS-sd-cts;

(2) $\psi_{sd}(A, \Delta)$ is SS-codense subset of $\tilde{V}$, for each SS-nowhere dense subset $(A, \Delta)$ of $\tilde{U}$.

**Proof**.

$(1) \Rightarrow (2)$ Let $(A, \Delta)$ is an SS-nowhere dense subset of $\tilde{U}$ and conversely assume that $int^s(\psi_{sd}(A, \Delta)) \neq \tilde{\varphi}$. It follows that,

there exists a non null SS-open subset $(K, \Lambda)$ of $\tilde{V}$ such that $(K, \Lambda) \tilde{\subseteq} \psi_{sd}(A, \Delta)$, and so

$$\psi_{sd}^{-1}(K, \Lambda) \tilde{\subseteq} \psi_{sd}^{-1}(\psi_{sd}(A, \Delta)).$$

Since $\psi_{sd}$ is injective, $\psi_{sd}^{-1}(K, \Lambda) \tilde{\subseteq} (A, \Delta)$, $\psi_{sd}^{-1}(K, \Lambda) \in SD(U)_{\Delta}$ from (1). Hence,

$$int^s(cl^s \psi_{sd}^{-1}(K, \Lambda)) \neq \tilde{\varphi}, \text{ which follows } int^s(cl^s(A, \Delta)) \neq \tilde{\varphi}.$$

Therefore,

$(A, \Delta)$ is not SS-nowhere dense subset of $\tilde{U}$, which contradicts our assumption.

Thus, $int^s(\psi_{sd}(A, \Delta)) = \tilde{\varphi}$. That is, $\psi_{sd}(A, \Delta)$ is SS-codense subset of $\tilde{V}$.

$(2) \Rightarrow (1)$ Let $(A, \Lambda) \in \sigma$. If $\psi_{sd}^{-1}(A, \Lambda) = \tilde{\varphi}$, then we get (1). If $\psi_{sd}^{-1}(A, \Lambda) \neq \tilde{\varphi}$. Assume conversely, $\psi_{sd}^{-1}(A, \Lambda) \notin SD(U)_{\Delta}$, which follows $\psi_{sd}^{-1}(A, \Lambda)$ is SS-nowhere dense subset of $\tilde{U}$. According to (2),

$(A, \Lambda) = \psi_{sd}(\psi_{sd}^{-1}(A, \Lambda))$ is SS-codense subset of $\tilde{V}$, given $\psi_{sd}$ is surjective; hence

$int^s(A, \Lambda) = (A, \Lambda) = \tilde{\varphi}$, which opposes our hypothesis. Therefore,

$\psi_{sd}^{-1}(A, \Lambda) \in SD(U)_{\Delta}$, and so we get (2).

**Remark 3.8** *The proof of Theorem 3.7 can not be hold in general without the bijectivity condition, as confirmed in the following example.*

**Example 3.9** *Let $U = \{p_1, p_2, p_3, p_4\}$, $V = \{q_1, q_2, q_3, q_4\}$, $\Delta = \{\gamma_1, \gamma_2\}$ and $\Lambda = \{\vartheta_1, \vartheta_2\}$.*

*Define $s : U \rightarrow V$ and $d : \Delta \rightarrow \Lambda$ as follows :*

$$s(p_1) = q_2, \ s(p_2) = q_2, \ s(p_3) = q_3, \ s(p_4) = q_3, \ d(\gamma_1) = \vartheta_1, \ d(\gamma_2) = \vartheta_2.$$

*Let $\tau = \{\tilde{U}, \tilde{\varphi}, (A_1, \Delta)\}$ be an STS over $U$ and*

$$\mu = \{\tilde{U}, \tilde{\varphi}, (A_i, \Delta), i = 1, 2, , \ldots, 7\}$$

*be an associated SSTS with $\tau$, where :*

$$A_1(\gamma_1) = \{p_1\}, \quad A_1(\gamma_2) = \varphi.$$

$$A_2(\gamma_1) = \{p_1, p_2\}, \quad A_2(\gamma_2) = \{p_1\}.$$

$$A_3(\gamma_1) = \{p_1, p_2\}, \quad A_3(\gamma_2) = \{p_3, p_4\}.$$

$$A_4(\gamma_1) = \{p_3, p_4\}, \quad A_4(\gamma_2) = \{p_1, p_2\}.$$

$$A_5(\gamma_1) = \{p_1, p_3, p_4\}, \quad A_5(\gamma_2) = \{p_1, p_2\}.$$

$$A_6(\gamma_1) = U, \quad A_6(\gamma_2) = \{p_1, p_2\}.$$

$$A_7(\gamma_1) = \{p_1, p_2\}, \quad A_7(\gamma_2) = \{p_1, p_3, p_4\}.$$

*Let $\sigma = \{\tilde{V}, \tilde{\varphi}, (B_1, \Lambda), (B_2, \Lambda)\}$ be an STS over $V$ and*

$$\mu^* = \{\tilde{V}, \tilde{\varphi}, (B_j, \Delta), j = 1, 2, \ldots, 5\}$$

*be an associated SSTS with σ, where:*

$$B_1(\vartheta_1) = \{q_2\}, \quad B_1(\vartheta_2) = \{q_3\}.$$

$$B_2(\vartheta_1) = \{q_1, q_2\}, \quad B_2(\vartheta_2) = \{q_1, q_3, q_4\}.$$

$$B_3(\vartheta_1) = \{q_1, q_2\}, \quad B_3(\vartheta_2) = \varphi.$$

$$B_4(\vartheta_1) = \{q_1, q_2\}, \quad B_4(\vartheta_2) = \{q_1\}.$$

$$B_5(\vartheta_1) = \{q_1\}, \quad B_5(\vartheta_2) = \{q_1\}.$$

*We have*

$$\psi_{sd}^{-1}(B_1, \Lambda) = \{(\vartheta_1, \{p_1, p_2\}), (\vartheta_2, \{p_3, p_4\})\} = \psi_{sd}^{-1}(B_2, \Lambda) \in SD(U)_\Delta$$

*Hence, $\psi_{sd}$ is an SS-sd-cts. On the other side, for the soft set $(C, \Delta)$, where:*

$C(\gamma_1) = \{p_2\}, C(\gamma_2) = \{p_3, p_4\}$, *is SS-nowhere dense subset of $\tilde{U}$, whereas $int^s[\psi_{sd}(C, \Delta)] = int^s[\{(\vartheta_1, \{q_2\}), (\vartheta_2, \{q_3\})\}] = \{(\vartheta_1, \{q_2\}), (\vartheta_2, \{q_3\})\} \neq \tilde{\varphi}$, which means $\psi_{sd}(C, \Delta)$ is not SS-codense subset of $\tilde{V}$. Hence, condition (2) in Theorem 3.7 is not hold because of it is not bijective.*

**Definition 3.10** *A soft map $\psi_{sd}$: $(U, \tau, \Delta) \to (V, \sigma, \Lambda)$ with $\tau \subset \mu$ and $\sigma \subset \mu^*$ is called:*

(1) *SS-semi\*-continuous (briefly, SS\*-semi-cts), if $\psi_{sd}^{-1}(G, \Lambda)$ is SS-semi-open subset of $\tilde{U}$ for each $(G, \Lambda) \in \mu^*$.*

(2) *SS-semi\*-open (briefly, SS-semi\*-open), if $\psi_{sd}(K, \Delta)$ is SS-semi-open subset of $\tilde{V}$ for each $(K, \Delta) \in \mu$.*

(3) *SS-\*-closed (briefly, SS\*-closed), if $\psi_{sd}(K, \Delta) \in \mu^{*c}$ for each $(K, \Delta) \in \mu^c$.*

**Theorem 3.11** *The following are equivalent for a bijection SS\*-closed map $\psi_{sd}$: $(U, \tau, \Delta) \to (V, \sigma, \Lambda)$ with $\tau \subset \mu$ and $\sigma \subset \mu^*$:*

(1) *$\psi_{sd}$ is SS-sd-cts;*

(2) *$\psi_{sd}(H, \Delta)$ is SS-nowhere dense subset of $\tilde{V}$, for each soft SS-nowhere dense and SS-closed subset $(H, \Delta)$ of $\tilde{U}$;*

(3) $\psi_{sd}^{-1}(K, \Lambda)$ *is SS-sd-subset of* $\tilde{U}$, *for each SS-sd-subset* $(K, \Lambda)$ *of* $\tilde{V}$.

**Proof**.

(1) $\Rightarrow$ (2) Assume that $(H, \Delta)$ is both an SS-nowhere dense and SS-closed subset of $\tilde{U}$. From Theorem 3.7, $\psi_{sd}(H, \Delta)$ is SS-codense subset of $\tilde{V}$ and so

$$int^s(cl^s(\psi_{sd}(H, \Delta))) = int^s(\psi_{sd}(H, \Delta)) = \tilde{\varphi}, \psi_{sd} \text{ is an SS*-closed.}$$

Therefore,

$$\psi_{sd}(H, \Delta) \text{ is SS-nowhere dense subset of } \tilde{V}.$$

(2) $\Rightarrow$ (3) Let $(K, \Lambda)$ is SS-sd-subset of $\tilde{V}$ and assume conversely $\psi_{sd}^{-1}(K, \Lambda)$ is not SS-sd-subset of $\tilde{U}$. Then,

$$int^s(cl^s(\psi_{sd}^{-1}(K, \Lambda))) = \tilde{\varphi}.$$

From (2),

$$int^s(cl^s(\psi_{sd}(\psi_{sd}^{-1}(K, \Lambda)))) = \tilde{\varphi}.$$

Since $\psi_{sd}$ is surjective, $int^s(cl^s(K, \Lambda)) = \tilde{\varphi}$ which contradicts our hypothesis. Therefore, we obtain (3).

(3) $\Rightarrow$ (1) Let $(G, \Lambda) \in \sigma$, then $(G, \Lambda)$ is SS-sd-subset of $\tilde{V}$. From (3), $\psi_{sd}^{-1}(G, \Lambda)$ is SS-sd-subset of $\tilde{U}$. Hence, $\psi_{sd}$ is SS-sd-cts.

**Theorem 3.12** *Let* $\psi_{sd}$: $(U, \tau, \Delta) \rightarrow (V, \sigma, \Lambda)$ *be an SS*-semi-open map with* $\tau \subset \mu$ *and* $\sigma \subset \mu^*$, *then the following are equivalent*:

(1) $\psi_{sd}$ *is an SS-sd-cts*;

(2) $int^s(\psi_{sd}(X, \Delta))$ *is SS-dense subset of* $\psi_{sd}(\tilde{V})$, *for each* $(X, \Delta)$ *which is both SS-open and SS-dense subset of* $\tilde{U}$.

**Proof**.

(1) $\Rightarrow$ (2) Let $(X, \Delta)$ is both SS-open and SS-dense subset of $\tilde{U}$, then from Theorem 3.5 (4), $\psi_{sd}(X, \Delta)$ is SS-dense subset of $\psi_{sd}(\tilde{U})$ which means

$$cl^s(\psi_{sd}(X, \Delta)) = \psi_{sd}(\tilde{U}).$$

Since $\psi_{sd}$ is an SS*-semi-open, $\psi_{sd}(X, \Delta)$ is SS-semi open subset of $\tilde{V}$, which follows

$\psi_{sd}(X, \Delta) \tilde{\subseteq} cl^s(int^s(\psi_{sd}(X, \Delta)))$. Hence, $\psi_{sd}(\tilde{U}) = cl^s(\psi_{sd}(X, \Delta)) \tilde{\subseteq} cl^s(int^s(\psi_{sd}(X, \Delta)))$.

However, we have

$$cl^s(int^s(\psi_{sd}(X, \Delta))) \tilde{\subseteq} \psi_{sd}(\tilde{U}).$$

Therefore,

$int^s(\psi_{sd}(X, \Delta))$ is SS-dense subset of $\psi_{sd}(\tilde{V})$.

(2) $\Rightarrow$ (1) Let $(X, \Lambda)$ is both SS-open and SS-dense subset of $\tilde{U}$. From (2),

$$\psi_{sd}(\tilde{V}) = cl^s(int^s(\psi_{sd}(X, \Delta))) \tilde{\subseteq} cl^s(\psi_{sd}(X, \Delta)).$$

However, we have

$$cl^s(\psi_{sd}(X, \Delta)) \tilde{\subseteq} \psi_{sd}(\tilde{V}).$$

Hence,

$$\psi_{sd}(X, \Delta) \text{ is SS-dense subset of } \psi_{sd}(\tilde{U}).$$

From Theorem 3.5 (1), $\psi_{sd}$ is an SS-sd-cts.

**Definition 3.13** [60] *A soft mapping $\psi_{sd}$: $(U, \tau, \Delta) \to (V, \sigma, \Lambda)$ with $\sigma \subset \mu^*$ is said to be:*

(1) *An SS-sd-open if $\psi_{sd}(G, \Delta) \in SD(V)_\Lambda$ for each non-null soft open subset $(G, \Delta)$ of $\tilde{U}$.*

(2) *An SS-sd-closed if either $\psi_{sd}(H, \Delta) \in SC(V)_\Lambda$ or $\psi_{sd}(H, \Delta) = \tilde{V}, \forall (H, \Delta) \in \tau^c$.*

**Theorem 3.14** *Let $\psi_{sd}$: $(U, \tau, \Delta) \to (V, \sigma, \Lambda)$ be a bijective soft map with $\sigma \subset \mu^*$, then $\psi_{sd}$ is an SS-sd-open iff it is an SS-sd-closed.*

**Proof. Necessity**: Let $(G, \Delta) \in \tau^c$, then $(G^{\tilde{c}}, \Delta) \in \tau$. Since $\psi_{sd}$ is bijective SS-sd-open,

$$\text{either } [\psi_{sd}(G, \Delta)]^{\tilde{c}} = \psi_{sd}(G^{\tilde{c}}, \Delta) \in SD(V)_\Lambda \text{ or } [\psi_{sd}(G, \Delta)]^{\tilde{c}} = \tilde{\varphi}.$$

It follows that,

$$\text{either } \psi_{sd}(G, \Delta) \in SC(V)_\Lambda \text{ or } \psi_{sd}(G, \Delta) = \tilde{V}.$$

Therefore, $\psi_{sd}$ is an SS-sd-closed.

**Sufficient**: By a similar way.

**Remark 3.15** *The proof of Theorem 3.14 can not be hold in general without the bijectivity condition, as confirmed in the following example.*

**Example 3.16** *Let $U = \{p_1, p_2, p_3, p_4\}$, $V = \{q_1, q_2, q_3, q_4\}$, $\Delta = \{\gamma_1, \gamma_2\}$ and $\Lambda = \{\vartheta_1, \vartheta_2\}$.*

*Define $s : U \to V$ and $d : \Delta \to \Lambda$ as follows :*

$$s(p_1) = q_1, \ s(p_2) = q_1, \ s(p_3) = q_1, \ s(p_4) = q_1, \ d(\gamma_1) = \vartheta_1, \ d(\gamma_2) = \vartheta_2.$$

*Let $\tau = \{\tilde{U}, \tilde{\varphi}, (A, \Delta)\}$ be an STS over $U$ , where :*

$$A(\gamma_1) = \{p_3\}, \quad A(\gamma_2) = \varphi.$$

*Let $\sigma = \{\tilde{V}, \tilde{\varphi}, (B_1, \Lambda)\}$ be an STS over $V$ and*

$$\mu^* = \{\tilde{V}, \tilde{\varphi}, (B_j, \Lambda), j = 1, 2, 3, 4\}$$

*be an associated SSTS with σ, where*:

$$B_1(\vartheta_1) = \{q_1, q_2, q_3\}, \quad B_1(\vartheta_2) = V.$$

$$B_2(\vartheta_1) = \{q_1, q_3\}, \quad B_2(\vartheta_2) = \varphi.$$

$$B_3(\vartheta_1) = \{q_1\}, \quad B_3(\vartheta_2) = \{q_1\}.$$

$$B_4(\vartheta_1) = \{q_1, q_3\}, \quad B_4(\vartheta_2) = \{q_1\}.$$

*then*

$$\psi_{sd}(A, \Delta) = \{(\vartheta_1, \{q_1\}), (\vartheta_2, \varphi)\}$$

*is an SS-sd-subset of $\tilde{V}$. On the other hand, we have*

$$\psi_{sd}(A^{\tilde{c}}, \Delta)] = \{(\vartheta_1, \{q_1\}), (\vartheta_2, \{q_1\})\}$$

*is not SS-sc-subset of $\tilde{V}$. It follows that, $\psi_{sd}$ is an SS-sd-open, but it is not SS-sd-closed, because it is not bijective.*

**Theorem 3.17** *The following are equivalent for a bijective soft map $\psi_{sd} \colon (U, \tau, \Delta) \to (V, \sigma, \Lambda)$ with $\sigma \subset \mu^*$:*

(1) *$\psi_{sd}$ is SS-sd-open;*

(2) *There exists a non-null SS-open subset $(B, \Lambda)$ of $\tilde{V}$ such that $(B, \Lambda) \tilde{\subseteq} cl^s[\psi_{sd}(A, \Delta)]$, for each non-null $(A, \Delta) \in \tau$;*

(3) *There exists a proper SS-closed subset $(D, \Lambda)$ of $\tilde{V}$ such that $int^s[\psi_{sd}(C, \Delta)] \tilde{\subseteq} (D, \Lambda)$, for each $(C, \Delta) \in \tau^c$ in which $\psi_{sd}(C, \Delta) \neq \tilde{V}$.*

**Proof**.

(1) $\Rightarrow$ (2) Let $\tilde{\varphi} \neq (A, \Delta) \in \tau$. From (1), $\psi_{sd}(A, \Delta) \in SD(V)_\Lambda$. Hence, there exists $\varphi \neq (B, \Lambda) \in \mu^*$ such that $(B, \Lambda) \tilde{\subseteq} cl^s(\psi_{sd}(B, \Delta))$, from Corollary 3.4.

(2) $\Rightarrow$ (3) Let $(C, \Delta) \in \tau^c$ such that $\psi_{sd}(C, \Delta) \neq \tilde{V}$, then $\tilde{\varphi} \neq (C^{\tilde{c}}, \Delta) \in \tau$. From (2), there exists a non-null SS-open subset $(D, \Lambda)$ of $\tilde{V}$ such that

$$(D, \Lambda) \tilde{\subseteq} cl^s[\psi_{sd}(C^{\tilde{c}}, \Delta)].$$

From Theorem 2.13 (1),

$$int^s(\psi_{sd}(C, \Delta)) = cl^s[\psi_{sd}(C^{\tilde{c}}, \Delta)]^{\tilde{c}} \tilde{\subseteq} (D^{\tilde{c}}, \Lambda),$$

$(D^{\tilde{c}}, \Lambda)$ is a proper SS-closed subset of $\tilde{V}$. Therefore, we obtain (3).

(3) $\Rightarrow$ (1) Let $\tilde{\varphi} \neq (C, \Delta) \in \tau$, then $\tilde{U} \neq (C^{\tilde{c}}, \Delta) \in \tau^c$. Since $\psi_{sd}$ is bijective, $\tilde{V} \neq \psi_{sd}(C^{\tilde{c}}, \Delta)$. From (3), there exists a proper SS-closed subset $(D, \Lambda)$ of $\tilde{V}$ such that

$$int^s[\psi_{sd}(C^{\tilde{c}}, \Delta)] \tilde{\subseteq} (D, \Lambda), \text{ which follows}$$

$$(D^{\tilde{c}}, \Lambda) \tilde{\subseteq} [int^s[\psi_{sd}(C^{\tilde{c}}, \Delta)]]^{\tilde{c}} = cl^s(\psi_{sd}(C, \Delta)), \text{ from Theorem 2.13 (1)}.$$

Therefore, $\psi_{sd}(C, \Delta) \in SD(V)_\Lambda$, from Corollary 3.4. Thus, $\psi_{sd}$ is an SS-sd-open.

**Remark 3.18** *The proof of Theorem 3.17 can not be hold in general without the bijectivity condition, as confirmed in the next example.*

**Example 3.19** *In Example 3.16, for the soft set* $(A, \Delta)$, *we have*

$$\psi_{sd}(A, \Delta) = \{(\vartheta_1, \{q_1\}), (\vartheta_2, \varphi)\}$$

*is an SS-sd-subset of* $\tilde{V}$ *which follows* $\psi_{sd}$ *is an SS-sd-open. On the other hand, we have*

$$int^s[\psi_{sd}(A, \Delta)] = \{(\vartheta_1, \{q_1\}), (\vartheta_2, \{q_1\})\}$$

*is not contained in any proper SS-closed subset of* $\tilde{V}$. *It follows that, condition (3) in Theorem 3.17 is not hold because it is not bijective.*

**Theorem 3.20** *Let* $\psi_{sd}$: $(U, \tau, \Delta) \to (V, \sigma, \Lambda)$ *be a soft map with* $\sigma \subset \mu^*$, *then the following are equivalent*:

(1) $\psi_{sd}$ *is SS-sd-closed*;

(2) *There exists a proper SS-closed subset* $(B, \Lambda)$ *of* $\tilde{V}$ *such that* $int^s[\psi_{sd}(A, \Delta)] \tilde{\subseteq} (B, \Lambda)$, *for each* $(A, \Delta) \in \tau^c$;

(3) *There exists a non-null SS-open subset* $(D, \Lambda)$ *of* $\tilde{V}$ *such that* $(D, \Lambda) \tilde{\subseteq} cl^s[\psi_{sd}(C, \Delta)]$, *for each* $(C, \Delta) \in \tau$ *in which* $\psi_{sd}(C, \Delta) \neq \tilde{V}$.

**Proof**.

$(1) \Rightarrow (2)$ Let $(A, \Delta) \in \tau^c$. From (1), $\psi_{sd}(A, \Delta) \in SC(V)_\Lambda$. Hence, there exists a proper SS-closed subset $(B, \Lambda)$ of $\tilde{V}$ such that $int^s[\psi_{sd}(A, \Delta)] \tilde{\subseteq} (B, \Lambda)$, from Corollary 3.4.

$(2) \Rightarrow (3)$ Let $(C, \Delta) \in \tau$ such that $\psi_{sd}(C, \Delta) \neq \tilde{V}$, then $(C^{\tilde{c}}, \Delta) \in \tau^c$. From (2), there exists a proper SS-closed subset $(D, \Lambda)$ of $\tilde{V}$ such that

$$int^s[\psi_{sd}(C^{\tilde{c}}, \Delta)] \tilde{\subseteq} (D, \Lambda).$$

From Theorem 2.13 (1),

$$(D^{\tilde{c}}, \Lambda) \tilde{\subseteq} [int^s[\psi_{sd}(C^{\tilde{c}}, \Delta)]]^{\tilde{c}} = cl^s(\psi_{sd}(C, \Delta)),$$

$(D^{\tilde{c}}, \Lambda)$ is a non-null SS-open subset of $\tilde{V}$. Therefore, we obtain (3).

$(3) \Rightarrow (1)$ Let $(C, \Delta) \in \tau^c$, then $(C^{\tilde{c}}, \Delta) \in \tau$. If $\psi_{sd}(C^{\tilde{c}}, \Delta) = \tilde{V}$, then we get the proof. If $\psi_{sd}(C^{\tilde{c}}, \Delta) \neq \tilde{V}$. By (3), there exists a non-null SS-open subset $(D, \Lambda)$ of $\tilde{V}$ such that

$$(D, \Lambda) \tilde{\subseteq} cl^s[\psi_{sd}(C^{\tilde{c}}, \Delta)].$$

From Theorem 2.13 (1),

$$int^s(\psi_{sd}(C, \Delta)) = [cl^s[\psi_{sd}(C^{\tilde{c}}, \Delta)]]^{\tilde{c}} \tilde{\subseteq} (D^{\tilde{c}}, \Lambda),$$

$(D^{\tilde{c}}, \Lambda)$ is a proper SS-closed subset of $\tilde{V}$. Therefore, $\psi_{sd}(C, \Delta) \in SC(V)_\Lambda$, from Corollary 3.4. Thus, $\psi_{sd}$ is SS-sd-closed.

**Proposition 3.21** *A soft subset* $(K, \Delta)$ *of an SSTS* $(U, \mu, \Delta)$ *is SS-dense iff its relative complement is SS-codense.*

**Proof**. Direct form Definition 3.6.

**Theorem 3.22** *Let* $\psi_{sd}$: $(U, \tau, \Delta) \to (V, \sigma, \Lambda)$ *be a soft map with* $\tau \subset \mu$ *and* $\sigma \subset \mu^*$, *then the following are equivalent*:

(1) $\psi_{sd}$ *is SS-sd-open*;

(2) $\psi_{sd}^{-1}(B, \Lambda)$ *is SS-dense subset of* $\tilde{U}$, *for each SS-open and SS-dense subset* $(B, \Lambda)$ *of* $\tilde{V}$;

(3) $\psi_{sd}^{-1}(B, \Lambda)$ *is SS-codense subset of* $\tilde{U}$, *for each SS-closed and SS-codense subset* $(B, \Lambda)$ *of* $\tilde{V}$.

**Proof**.

(1) $\Rightarrow$ (2) Let $(B, \Lambda)$ is SS-open and SS-dense subset of $\tilde{V}$ and assume conversely $cl^s(\psi_{sd}^{-1}(B, \Lambda)) \neq \tilde{U}$, which follows there exists a proper SS-closed subset $(H, \Lambda)$ of $\tilde{V}$ such that

$$\psi_{sd}^{-1}(B, \Lambda) \tilde{\subseteq} (H, \Lambda), \text{ and so}$$

$$(H^{\tilde{c}}, \Lambda) \tilde{\subseteq} \psi_{sd}^{-1}(B^{\tilde{c}}, \Lambda). \text{ It follows that,}$$

$$cl^s(\psi_{sd}(H^{\tilde{c}}, \Lambda)) \tilde{\subseteq} cl^s(\psi_{sd}(\psi_{sd}^{-1}(B^{\tilde{c}}, \Lambda))) \tilde{\subseteq} cl^s(B^{\tilde{c}}, \Lambda) = (B^{\tilde{c}}, \Lambda) \tag{3}$$

Since $(H^{\tilde{c}}, \Lambda)$ is non-null SS-open set, $\psi_{sd}(H^{\tilde{c}}, \Lambda) \in SD(V)_\Lambda$, from (1). That is, there exists $\tilde{\varphi} \neq (G, \Lambda) \in \mu^*$ such that

$$(G, \Lambda) \tilde{\subseteq} cl^s(\psi_{sd}(H^{\tilde{c}}, \Lambda)) \tag{4}$$

From Eqs (3) and (4), $(G, \Lambda) \tilde{\subseteq} (B^{\tilde{c}}, \Lambda)$, which follows $(B, \Lambda) \tilde{\subseteq} (G^{\tilde{c}}, \Lambda)$, $(G^{\tilde{c}}, \Lambda)$ is a proper SS-closed set. Hence, $cl^s(B, \Lambda) \neq \tilde{V}$, which contradicts our hypothesis. Therefore,

$$\psi_{sd}^{-1}(B, \Lambda) \text{ is SS-dense subset of } \tilde{U}.$$

(2) $\Rightarrow$ (3) Let $(B, \Lambda)$ is SS-closed and SS-codense subset of $\tilde{V}$, then $(B^{\tilde{c}}, \Lambda)$ is SS-open and SS-dense set, from Proposition 3.21. Hence, $\psi_{sd}^{-1}(B^{\tilde{c}}, \Lambda)$ is SS-dense subset of $\tilde{U}$, from (2). That is,

$$cl^s[\psi_{sd}^{-1}(B^{\tilde{c}}, \Lambda)] = \tilde{U}, \text{ which follows}$$

$$int^s(\psi_{sd}^{-1}(B, \Lambda)) = (cl^s[\psi_{sd}^{-1}(B^{\tilde{c}}, \Lambda)])^{\tilde{c}} = [\tilde{U}]^{\tilde{c}} = \tilde{\varphi}.$$

Therefore,

$$\psi_{sd}^{-1}(B, \Lambda) \text{ is SS-codense subset of } \tilde{U}.$$

(3) $\Rightarrow$ (1) Let $\tilde{\varphi} \neq (A, \Delta) \in \tau$ and conversely assume that $\psi_{sd}(A, \Delta) \notin SD(V)_\Lambda$, which follows $int^s(cl^s(\psi_{sd}(A, \Delta))) = \tilde{\varphi}$. Hence, $cl^s(\psi_{sd}(A, \Delta))$ is SS-closed and SS-codense set. By (3),

$$int^s(\psi_{sd}^{-1}(cl^s(\psi_{sd}(A, \Delta)))) = \tilde{\varphi}. \text{ It follows that,}$$

$$int^s(A, \Delta) = \tilde{\varphi}, \text{ which contradicts our hypothesis.}$$

Therefore, $\psi_{sd}(A, \Delta) \in SD(V)_\Lambda$, and hence $\psi_{sd}$ is an SS-sd-open.

**Corollary 3.23** *Let* $\psi_{sd}: (U, \tau, \Delta) \to (V, \sigma, \Lambda)$ *be an SS\*-semi cts map with* $\tau \subset \mu$ *and* $\sigma \subset \mu^*$, *then the following are equivalent*:

(1) $\psi_{sd}$ *is SS-sd-open*;

(2) $int^s(\psi_{sd}^{-1}(B, \Lambda))$ *is SS-dense subset of* $\tilde{U}$, *for each SS-open and SS-dense subset* $(B, \Lambda)$ *of* $\tilde{V}$.

**Proof**.

(1) $\Rightarrow$ (2) Let $(B, \Lambda)$ is SS-open and SS-dense subset of $\tilde{V}$, then

$$cl^s(\psi_{sd}^{-1}(B, \Lambda)) = \tilde{U}, \; from \; Theorem \; 3.22. \tag{5}$$

Since $\psi_{sd}$ is an SS*-semi cts, $\psi_{sd}^{-1}(B, \Lambda)$ is SS-semi-open. That is,

$\psi_{sd}^{-1}(B, \Lambda) \tilde{\subseteq} cl^s(int^s(\psi_{sd}^{-1}(B, \Lambda)))$, which implies

$$cl^s(\psi_{sd}^{-1}(B, \Lambda)) \tilde{\subseteq} cl^s(int^s(\psi_{sd}^{-1}(B, \Lambda))) \tag{6}$$

From Eqs (5) and (6), $cl^s(int^s(\psi_{sd}^{-1}(B, \Lambda))) = \tilde{U}$. Thus, $int^s(\psi_{sd}^{-1}(B, \Lambda))$ is SS-dense subset of $\tilde{U}$.

(2) $\Rightarrow$ (1) Clear from Theorem 3.22.

**Definition 3.24** *A soft mapping* $\psi_{sd}$: $(U, \tau, \Delta) \rightarrow (V, \sigma, \Lambda)$ *with* $\tau \subset \mu$ *and* $\sigma \subset \mu^*$ *is said to be* SS*-sd-open if*

$$\psi_{sd}(G, \Delta) \in SD(V)_\Lambda \; for \; each \; \tilde{\varphi} \neq (G, \Delta) \in SD(U)_\Delta.$$

**Theorem 3.25** *Let* $\psi_{sd}$: $(U, \tau, \Delta) \rightarrow (V, \sigma, \Lambda)$ *be an SS*-semi cts map with* $\tau \subset \mu$ *and* $\sigma \subset \mu^*$, *then the following are equivalent*:

(1)  $\psi_{sd}$ *is* SS-sd-open;

(2)  $\psi_{sd}^{-1}(B, \Lambda)$ *is SS-nowhere dense subset of* $\tilde{U}$, *for each SS-closed and SS-codense subset* $(B, \Lambda)$ *of* $\tilde{V}$;

(3)  $\psi_{sd}$ *is* SS*-sd-open*.

**Proof**.

(1) $\Rightarrow$ (2) Consider $(B, \Lambda)$ be an SS-closed and SS-codense subset of $\tilde{V}$, then

$$int^s(\psi_{sd}^{-1}(B, \Lambda)) = \tilde{\varphi}, \; from \; Theorem \; 3.22. \tag{7}$$

Since $\psi_{sd}$ is an SS*-semi cts, $\psi_{sd}^{-1}(B, \Lambda)$ is SS-semi-closed subset of $\tilde{V}$.
Hence,

$$int^s(cl^s(\psi_{sd}^{-1}(B, \Lambda))) \tilde{\subseteq} \psi_{sd}^{-1}(B, \Lambda), \; \text{which follows}$$

$$int^s(cl^s(\psi_{sd}^{-1}(B, \Lambda))) \tilde{\subseteq} int^s(\psi_{sd}^{-1}(B, \Lambda)) = \tilde{\varphi}, \text{from Eq (7)}.$$

Therefore, $int^s(cl^s(\psi_{sd}^{-1}(B, \Lambda))) = \tilde{\varphi}$. Thus, $\psi_{sd}^{-1}(B, \Lambda)$ is SS-nowhere dense subset of $\tilde{U}$.

(2) $\Rightarrow$ (3) Let $(G, \Delta) \in SD(U)_\Delta$ and assume conversely $\psi_{sd}(G, \Delta) \notin SD(V)_\Lambda$, which implies $int^s(cl^s(\psi_{sd}(G, \Delta))) = \tilde{\varphi}$. Hence, $cl^s(\psi_{sd}(G, \Delta))$ is SS-closed and SS-codense subset of $\tilde{V}$. From (2), $\psi_{sd}^{-1}[cl^s(\psi_{sd}(G, \Delta))]$ is SS-nowhere dense subset of $\tilde{U}$, which means

$$int^s(cl^s(\psi_{sd}^{-1}[cl^s(\psi_{sd}(G, \Delta))])) = \tilde{\varphi}.$$

It is owing,

$int^s(cl^s(G, \Delta)) \tilde{\subseteq} int^s(cl^s(\psi_{sd}^{-1}[cl^s(\psi_{sd}(G, \Delta))])) = \tilde{\varphi}$, from Theorem 2.5 (3).

Therefore, $(G, \Delta) \notin SD(U)_\Delta$, which contradicts our hypothesis. Thus, $\psi_{sd}(G, \Delta) \in SD(V)_\Delta$, and hence $\psi_{sd}$ is an SS*-sd-open.

$(3) \Rightarrow (1)$ Let $(G, \Delta) \in \tau$, then $(G, \Delta) \in SD(U)_\Delta$. By (3), $\psi_{sd}(G, \Delta) \in SD(V)_\Delta$, which follows $\psi_{sd}$ is an SS-sd-open.

**Remark 3.26** *If $\psi_{sd}$ is not SS\*-semi cts in Theorem 3.25, then the proof can not be hold in general, as declared in the next example.*

**Example 3.27** *Let $U = \{p_1, p_2, p_3, p_4\}$, $V = \{q_1, q_2, q_3, q_4\}$, $\Delta = \{\gamma_1, \gamma_2\}$ and $\Lambda = \{\vartheta_1, \vartheta_2\}$.*

*Define $s : U \rightarrow V$ and $d : \Delta \rightarrow \Lambda$ as follows :*

$$s(p_1) = q_4, \ s(p_2) = q_4, \ s(p_3) = q_4, \ s(p_4) = q_4, \ d(\gamma_1) = \vartheta_1, \ d(\gamma_2) = \vartheta_2.$$

*Let $\tau = \{\tilde{U}, \tilde{\varphi}, (H_1, \Delta)\}$ be an STS over U and*

$$\mu = \{\tilde{U}, \tilde{\varphi}, (H_i, \Delta), i = 1, 2, 3, 4\}$$

*be an associated SSTS with $\tau$, where:*

$$H_1(\gamma_1) = \{p_3\}, \quad H_1(\gamma_2) = \varphi.$$

$$H_2(\gamma_1) = \{p_1, p_3\}, \quad H_2(\gamma_2) = \{p_2\}.$$

$$H_3(\gamma_1) = \{p_1, p_3\}, \quad H_3(\gamma_2) = \varphi.$$

$$H_4(\gamma_1) = \{p_1\}, \quad H_4(\gamma_2) = \{p_1\}.$$

*Let $\sigma = \{\tilde{V}, \tilde{\varphi}, (K_1, \Lambda)\}$ be an STS over V and*

$$\mu^* = \{\tilde{V}, \tilde{\varphi}, (K_j, \Lambda), j = 1, 2, .., 5\}$$

*be an associated SSTS with $\sigma$, where:*

$$K_1(\vartheta_1) = \{q_1, q_2, q_3\}, \quad K_1(\vartheta_2) = V.$$

$$K_2(\vartheta_1) = \{q_3, q_4\}, \quad K_2(\vartheta_2) = \varphi.$$

$$K_3(\vartheta_1) = \{q_4\}, \quad K_3(\vartheta_2) = \{q_4\}.$$

$$K_4(\vartheta_1) = \{q_3, q_4\}, \quad K_4(\vartheta_2) = \{q_4\}.$$

$$K_5(\vartheta_1) = \{q_4\}, \quad K_5(\vartheta_2) = \varphi.$$

*Then,*

$$\psi_{sd}^{-1}(K_1, \Lambda) = \{(\gamma_1, \varphi), (\gamma_2, U)\} \text{ and}$$

$$cl^s(int^s(\psi_{sd}^{-1}(K_1, \Lambda))) = \tilde{\varphi} \text{ and hence}$$

*$\psi_{sd}^{-1}(K_1, \Lambda)$ is not SS-semi-open subset of $\tilde{U}$, which follows $\psi_{sd}$ is not SS\*-semi cts.*
*Also, $\psi_{sd}(H_1, \Delta) = \{(\vartheta_1, \{q_4\}), (\vartheta_2, \varphi)\} \in SD(V)_\Lambda$, which implies $\psi_{sd}$ is an SS-sd-open.*

*Moreover, for each* $(G, \Delta) \in SD(U)_\Delta$, *we have*

$$\psi_{sd}(G, \Delta) \in \{\{(\vartheta_1, \{q_4\}), (\vartheta_2, \varphi)\}, \{(\vartheta_1, \{q_4\}), (\vartheta_2, \{q_4\})\}, \{(\vartheta_1, \varphi), (\vartheta_2, \{q_4\})\}\} \in SD(V)_\Lambda,$$

*which implies* $\psi_{sd}$ *is an SS\*-sd-open.*

*On the other hand, for the soft set* $(Z, \Lambda) = \{(\vartheta_1, \{q_1, q_2\}), (\vartheta_2, V)\}$ *which is SS-closed and SS-codense subset of* $\tilde{V}$, *we have*

$$int^s(cl^s(\psi_{sd}^{-1}(Z, \Lambda))) = int^s(cl^s(\{(\vartheta_1, \varphi), (\vartheta_2, U)\})) = \{(\gamma_1, \{q_1\}), (\gamma_2, \{q_1\})\} \neq \tilde{\varphi}, and$$
*hence*

$\psi_{sd}^{-1}(Z, \Lambda)$ *is not SS-nowhere dense subset of* $\tilde{U}$. *Therefore,* $\psi_{sd}$ *satisfies conditions (1) and (3), but not (2) in Theorem 3.25, because it is not SS\*-semi cts.*

## 4 New version of connectedness via supra soft sd-sets

This section is devoted to introduce the concept of SS-sd-separated sets as a prelude to introducing the connectedness in SSTS via SS-sd-sets, named SS-sd-connectedness. We show that, this concept is general than the earlier studies. We prove that the pre-image of an SS-sd-separated sets under a surjective SS-sd-irresolute map is an SS-sd-separated. Moreover, we prove that, there is no priori relationship between SS-sd-connectedness in an SSTS and its parametric supra topological spaces in general, supported by concrete counterexamples. Finally, we prove that the image of an SS-sd-connected set under an SS-sd-irresolute map is an SS-sd-connected.

**Definition 4.1** [56] *Any pair of non-null soft subsets* $(G, \Delta)$, $(H, \Delta)$ *of an SSTS* $(U, \mu, \Delta)$ *are said to be SS-separated if* $(G, \Delta) \tilde{\cap} cl^s(H, \Delta) = \tilde{\varphi}$ *and* $cl^s(G, \Delta) \tilde{\cap} ((H, \Delta)) = \tilde{\varphi}$.

**Definition 4.2** Any pair of non-null soft subsets $(G, \Delta)$, $(H, \Delta)$ of an SSTS $(U, \mu, \Delta)$ are said to be SS-sd-separated if $(G, \Delta) \tilde{\cap} cl^s_{sd}(H, \Delta) = \tilde{\varphi}$ and $cl^s_{sd}(G, \Delta) \tilde{\cap} ((H, \Delta)) = \tilde{\varphi}$.

**Lemma 4.3** Every pair of SS-sd-separated subsets $(G, \Delta)$ and $(H, \Delta)$ of an SSTS $(U, \mu, \Delta)$ is disjoint, but not conversely hold in general, as declared in the next example.

**Example 4.4** *Let* $V = \{i_1, i_2, i_3, i_4\}$, $\Delta = \{\vartheta_1, \vartheta_2\}$ *and* $\mu = \{\tilde{V}, \tilde{\varphi}, (L_j, \Delta), j = 1, 2, .., 5\}$ *be an SSTS on V, where*:

$$L_1(\vartheta_1) = \{i_4\}, \quad L_1(\vartheta_2) = \varphi.$$

$$L_2(\vartheta_1) = \{i_3, i_4\}, \quad L_2(\vartheta_2) = \varphi.$$

$$L_3(\vartheta_1) = \{i_4\}, \quad L_3(\vartheta_2) = \{i_4\}.$$

$$L_4(\vartheta_1) = \{i_3, i_4\}, \quad L_4(\vartheta_2) = \{i_4\}.$$

$$L_5(\vartheta_1) = \{i_1, i_2, i_3\}, \quad L_5(\vartheta_2) = V.$$

*For the soft sets* $(T, \Delta)$ *and* $(S, \Delta)$, *where*:

$$T(\vartheta_1) = \{i_1, i_2\}, \quad T(\vartheta_2) = V.$$

$$S(\vartheta_1) = \{i_3, i_4\}, \quad S(\vartheta_2) = \varphi.$$

*We have that* $(T, \Delta)$ *and* $(S, \Delta)$ *are disjoint. However, they are not SS-sd-separated, since* $(T, \Delta) \tilde{\cap} cl^s_{sd}(S, \Delta) \neq \tilde{\varphi}$.

**Note 4.5** The reverse inclusion in Lemma 4.3 can be held in case of $(G, \Delta)$ and $(H, \Delta)$ are SS-sc-sets or SS-sd-sets together.

The following propositions characterize the SS-sd-separated sets and follows from Definition 4.2. So, the proof is omitted and left to the reader.

**Proposition 4.6** Let $(G, \Delta)$ and $(H, \Delta)$ be soft subsets an SSTS $(U, \mu, \Delta)$, then the following results are hold:

(1) If $(G, \Delta)$ and $(H, \Delta)$ are SS-sd-separated such that $(A, \Delta)\tilde{\subseteq}(G, \Delta)$ and $(B, \Delta)\tilde{\subseteq}(H, \Delta)$, then $(A, \Delta)$ and $(B, \Delta)$ are also SS-sd-separated.

(2) If $(G, \Delta)$ and $(H, \Delta)$ are SS-sd-sets such that $(A, \Delta) = (G, \Delta)\tilde{\cap}(H^{\tilde{c}}, \Delta)$ and $(B, \Delta) = (H, \Delta)\tilde{\cap}(G^{\tilde{c}}, \Delta)$, then $(A, \Delta)$ and $(B, \Delta)$ are SS-sd-separated.

**Proposition 4.7** A pair of soft subsets $(G, \Delta)$ and $(H, \Delta)$ of an SSTS $(U, \mu, \Delta)$ is SS-sd-separated iff there are SS-sd-sets $(A, \Delta)$ and $(B, \Delta)$ such that $(G, \Delta)\tilde{\subseteq}(A, \Delta)$, $(H, \Delta)\tilde{\subseteq}(B, \Delta)$, $(G, \Delta)\tilde{\cap}(B, \Delta) = \tilde{\varphi}$ and $(H, \Delta)\tilde{\cap}(A, \Delta) = \tilde{\varphi}$.

**Proposition 4.8** Every pair of SS-separated sets is SS-sd-separated, but not conversely in general, as declared in the next example.

**Example 4.9** Let $U = \{j_1, j_2, j_3\}$, $\Delta = \{\gamma_1, \gamma_2\}$ and $\mu = \{\tilde{U}, \tilde{\varphi}, (S_i, \Delta), i = 1, 2, 3, 4\}$ be an SSTS on U, where:

$$S_1(\gamma_1) = \{j_1\}, \quad S_1(\gamma_2) = \varphi.$$

$$S_2(\gamma_1) = \{j_1\}, \quad S_2(\gamma_2) = \{j_1\}.$$

$$S_3(\gamma_1) = \{j_1, j_2\}, \quad S_3(\gamma_2) = \{j_1, j_3\}.$$

$$S_4(\gamma_1) = \{j_2\}, \quad S_4(\gamma_2) = \{j_1, j_3\}.$$

For the soft sets $(M, \Delta)$ and $(N, \Delta)$, where:

$$M(\gamma_1) = \{j_2, j_3\}, \quad M(\gamma_2) = \{j_1, j_3\}.$$

$$N(\gamma_1) = \{j_1\}, \quad N(\gamma_2) = \{j_2\}.$$

We have that $(M, \Delta)$ and $(N, \Delta)$ are SS-sd-separated sets, but not SS-separated.

**Definition 4.10** [60] A soft map $\psi_{sd}$: $(U, \tau, \Delta) \to (V, \sigma, \Lambda)$ with $\tau \subset \mu$ and $\sigma \subset \mu^*$, is said to be SS-sd-irresolute if either $\psi_{sd}^{-1}(Y, \Lambda) = \tilde{\varphi}$ or $\psi_{sd}^{-1}(Y, \Lambda) \in SD(U)_\Delta$ for each $(Y, \Lambda) \in SD(V)_\Lambda$.

**Theorem 4.11** [60] Let $\psi_{sd}$: $(U, \tau, \Delta) \to (V, \sigma, \Lambda)$ be a soft map with $\tau \subset \mu$ and $\sigma \subset \mu^*$, then the following are equivalent:

(1) $\psi_{sd}$ is SS-sd-irresolute.

(2) For each $(L, \Lambda) \in SC(V)_\Lambda$, either $\psi_{sd}^{-1}(L, \Lambda) \in SC(U)_\Delta$ or $\psi_{sd}^{-1}(L, \Lambda) = \tilde{U}$.

(3) $cl_{sd}^s(\psi_{sd}^{-1}(L, \Lambda))\tilde{\subseteq}\psi_{sd}^{-1}(cl_{sd}^s(L, \Lambda)) \; \forall \; (L, \Lambda)\tilde{\subseteq}\tilde{V}$

(4) $\psi_{sd}(cl_{sd}^s(M, \Delta))\tilde{\subseteq}cl_{sd}^s(\psi_{sd}(M, \Delta)) \; \forall \; (M, \Delta)\tilde{\subseteq}\tilde{U}.$

(5) $\psi_{sd}^{-1}(int_{sd}^s(L, \Lambda))\tilde{\subseteq}int_{sd}^s(\psi_{sd}^{-1}(L, \Lambda)) \; \forall \; (L, \Lambda)\tilde{\subseteq}\tilde{V}.$

**Proposition 4.12** The pre-image of SS-sd-separated sets under a surjective SS-sd-irresolute map is SS-sd-separated.

**Proof**. Let $\psi_{sd}$: $(U, \tau, \Delta) \to (V, \sigma, \Lambda)$ is a surjective SS-sd-irresolute map with $\tau \subset \mu$ and $\sigma \subset \mu^*$ such that $(Y, \Lambda)$ and $(Z, \Lambda)$ are SS-sd-separated subsets of $\tilde{V}$. Since $cl_{sd}^s(\psi_{sd}^{-1}(Y, \Lambda)) \tilde{\subseteq} \psi_{sd}^{-1}(cl_{sd}^s(Y, \Lambda)) \; \forall \; (Y, \Lambda) \in S(V)_\Lambda)$, from Theorem 4.11 (3).

Hence,

$$\psi_{sd}^{-1}(Z, \Lambda) \tilde{\cap} cl_{sd}^s(\psi_{sd}^{-1}(Y, \Lambda)) \tilde{\subseteq} \psi_{sd}^{-1}(Z, \Lambda) \tilde{\cap} \psi_{sd}^{-1}(cl_{sd}^s(Y, \Lambda))$$

$$= \psi_{sd}^{-1}[(Z, \Lambda) \tilde{\cap} cl_{sd}^s(Y, \Lambda)]$$

$$= \psi_{sd}^{-1}(\tilde{\varphi}) = \tilde{\varphi}, \; \psi_{sd} \text{ is surjective.}$$

Follows a similar argument, one can get

$$\psi_{sd}^{-1}(Y, \Lambda) \tilde{\cap} cl_{sd}^s(\psi_{sd}^{-1}(Z, \Lambda)) = \tilde{\varphi}.$$

Therefore,

$$\psi_{sd}^{-1}(Y, \Lambda), \psi_{sd}^{-1}(Z, \Lambda) \text{ are SS-sd-separated subsets of } \tilde{U}.$$

**Definition 4.13** [57] *An SSTS $(U, \mu, \Delta)$ is said to be SS-connected, if $\tilde{U}$ can not be written as a soft union of any non-null SS-separated subsets $(E, \Delta)$, $(R, \Delta)$ of $\tilde{U}$. Otherwise, $(U, \mu, \Delta)$ is said to be SS-sd-disconnected.*

**Definition 4.14** An SSTS $(U, \mu, \Delta)$ is said to be SS-sd-connected, if $\tilde{U}$ can not be written as a soft union of any non-null SS-sd-separated subsets $(E, \Delta)$, $(R, \Delta)$ of $\tilde{U}$. In this case $(E, \Delta)$ and $(R, \Delta)$ are called SS-sd-disconnection of $\tilde{U}$. Otherwise, $(U, \mu, \Delta)$ is said to be SS-sd-disconnected. A soft subset $(Y, \Delta)$ of $(U, \mu, \Delta)$ is SS-sd-connected if it is SS-sd-connected subspace of $\tilde{U}$.

**Theorem 4.15** Let $(N, \mu, \Delta)$ be an SSTS, then the following properties are equivalent:

(1) $\tilde{N}$ is an SS-sd-connected.

(2) There are no disjoint SS-sd-sets $(E, \Delta)$ and $(R, \Delta)$ in which $\tilde{N} = (E, \Delta) \tilde{\cup} (R, \Delta)$.

(3) There are no disjoint SS-sc-sets $(E, \Delta)$ and $(R, \Delta)$ in which $\tilde{N} = (E, \Delta) \tilde{\cup} (R, \Delta)$.

(4) There is no proper soft subset of $\tilde{N}$ which is both SS-sc-set and SS-sd-set.

(5) There are no SS-sd-separated sets $(E, \Delta)$ and $(R, \Delta)$ in which $\tilde{N} = (E, \Delta) \tilde{\cup} (R, \Delta)$.

**Proof**.

(1) $\Rightarrow$ (2) Suppose conversely, $\tilde{N} = (E, \Delta) \tilde{\cup} (R, \Delta)$ for some disjoint SS-sd-subsets of $\tilde{N}$, then $(E, \Delta) = (R^{\tilde{c}}, \Delta)$ is SS-sc-set. It follows that, $cl_{sd}^s(R^{\tilde{c}}, \Delta) = (R^{\tilde{c}}, \Delta)$. Hence, $\tilde{N} = (R^{\tilde{c}}, \Delta) \tilde{\cup} (R, \Delta)$ whereas $(R^{\tilde{c}}, \Delta)$ and $(R, \Delta)$ are two non-null SS-sd-separated sets, which contradicts (1).

(2) $\Rightarrow$ (3) Assume conversely that $\tilde{N} = (E, \Delta) \tilde{\cup} (R, \Delta)$ for some disjoint SS-sc-sets $(E, \Delta)$ and $(R, \Delta)$, then $(E, \Delta)$ and $(R, \Delta)$ are SS-sd-sets, which contradicts (2).

(3) $\Rightarrow$ (4) Suppose conversely that there is a proper soft subset $(E, \Delta)$ of $\tilde{N}$ which is both SS-sc-set and SS-sd-set. It follows that, $\tilde{N} = (E^{\tilde{c}}, \Delta) \tilde{\cup} (E, \Delta)$ and $(E^{\tilde{c}}, \Delta) \tilde{\cap} (E, \Delta) = \tilde{\varphi}$, which contradicts (3).

(4) $\Rightarrow$ (3) Suppose conversely that $\tilde{N} = (E, \Delta) \tilde{\cup} (R, \Delta)$ for some disjoint SS-sc-sets $(E, \Delta)$ and $(R, \Delta)$, then $(E, \Delta) = (R^{\tilde{c}}, \Delta)$ and $(R^{\tilde{c}}, \Delta) = (E, \Delta)$, which contradicts (4).

$(3) \Rightarrow (5)$ Suppose conversely that $\tilde{N} = (E, \Delta)\tilde{\cup}(R, \Delta)$ for some SS-sd-separated sets $(E, \Delta)$ and $(R, \Delta)$, then $(R, \Delta)\tilde{\cap}cl_{sd}^s(E, \Delta) = \tilde{\varphi}$ and $cl_{sd}^s(R, \Delta)\tilde{\cap}(E, \Delta) = \tilde{\varphi}$. So, $(E, \Delta)\tilde{\cap}(R, \Delta) = \tilde{\varphi}$. Hence, $(E, \Delta) = (R^{\tilde{c}}, \Delta)$ and $(R, \Delta) = (E^{\tilde{c}}, \Delta)$. Therefore, $cl_{sd}^s(E, \Delta)\tilde{\subseteq}(R^{\tilde{c}}, \Delta) = (E, \Delta)$ and $cl_{sd}^s(R, \Delta)\tilde{\subseteq}(E^{\tilde{c}}, \Delta) = (R, \Delta)$. Therefore, $(E, \Delta)$ and $(R, \Delta)$ are SS-sc-sets, which contradicts (3).

$(5) \Rightarrow (1)$ Follows from Definition 4.14.

**Corollary 4.16 (1)** *Every SS-disconnected space is SS-sd-disconnected.*

**(2)** *Every coarser SSTS from an SS-sd-connected spaces is SS-sd-connected.*

**Proof**. Clear from Theorem 4.15.

**Remark 4.17** *The converse of Corollary 4.16 is not true in general, as confirmed in the next examples.*

**Examples 4.18 (1)** *In Example 4.9, we have $\tilde{U} = (M, \Delta)\tilde{\cup}(N, \Delta)$ whereas $(M, \Delta), (N, \Delta)$ are SS-sd-separated sets. Hence, $(U, \mu, \Delta)$ is SS-sd-disconnected space. Also, it easy to check that $\tilde{U}$ is an SS-connected.*

**(2)** *Let $U = \{f_1, f_2\}, \Delta = \{\gamma_1, \gamma_2\}$ and consider the soft sets $(Z_i, \Delta), i = 1, 2, \ldots, 7$ over U, where:*

$$Z_1(\gamma_1) = \{f_1\}, \quad Z_1(\gamma_2) = \varphi.$$

$$Z_2(\gamma_1) = \{f_1\}, \quad Z_2(\gamma_2) = \{f_1\}.$$

$$Z_3(\gamma_1) = \{f_1\}, \quad Z_3(\gamma_2) = U.$$

$$Z_4(\gamma_1) = U, \quad Z_4(\gamma_2) = \varphi.$$

$$Z_5(\gamma_1) = U, \quad Z_5(\gamma_2) = \{f_2\}.$$

$$Z_6(\gamma_1) = U, \quad Z_6(\gamma_2) = \{f_1\}.$$

$$Z_7(\gamma_1) = \varphi, \quad Z_7(\gamma_2) = \{f_1\}.$$

*Consider $\mu = \{\tilde{U}, \tilde{\varphi}, (Z_i, \Delta), i = 1, 2, \ldots, 6\}$ be an SSTS on U, then $(U, \mu, \Delta)$ is SS-sd-connected. On the other side, consider $\mu \subset \mu^*$, where $\mu^* = \{\tilde{U}, \tilde{\varphi}, (Z_i, \Delta), i = 1, 2, \ldots, 7\}$. Then, $\tilde{U} = (A, \Delta)\tilde{\cup}(B, \Delta)$ whereas $(A, \Delta), (B, \Delta)$ are disjoint SS-sd-sets, where:*

$$A(\gamma_1) = U, \quad Z_1(\gamma_2) = \varphi.$$

$$Z_2(\gamma_1) = \varphi, \quad Z_2(\gamma_2) = U.$$

*Therefore, $(U, \mu^*, \Delta)$ is an SS-sd-disconnected.*

**Corollary 4.19** *Let $(N, \mu, \Delta)$ be an SSTS, then the following properties are equivalent:*

(1) *$\tilde{N}$ is SS-sd-connected.*

(2) *If $\tilde{N}$ can be written as a soft union of two disjoint SS-sd-sets, then one of them is the null soft set.*

(3) *If $\tilde{N}$ can be written as a soft union of two disjoint SS-sc-sets, then one of them is the null soft set.*

**Proof**. Direct from Theorem 4.15.

**Remark 4.20** *There is no priori relationship between an SS-sd-connected SSTS and its parametric supra topological spaces in general, as shown in the following examples.*

**Examples 4.21 (1)** *Let $U = \{f_1, f_2, f_3\}$, $\Delta = \{\gamma_1, \gamma_2\}$ and $\mu = \{\tilde{U}, \tilde{\varphi}, (X_i, \Delta), i = 1, 2, .., 5\}$ be an SSTS on U, where:*

$$X_1(\gamma_1) = \{f_1, f_2\}, \quad X_1(\gamma_2) = \{f_1, f_3\}.$$

$$X_2(\gamma_1) = \{f_2, f_3\}, \quad X_2(\gamma_2) = \{f_2, f_3\}.$$

$$X_3(\gamma_1) = \{f_1, f_3\}, \quad X_3(\gamma_2) = \{f_1, f_3\}.$$

$$X_4(\gamma_1) = \{f_2, f_3\}, \quad X_4(\gamma_2) = U.$$

$$X_5(\gamma_1) = U, \quad X_5(\gamma_2) = \{f_1, f_3\}.$$

*For the soft sets $(A, \Delta)$, $(B, \Delta)$, where:*

$$A(\gamma_1) = U, \quad A(\gamma_2) = \varphi.$$

$$B(\gamma_1) = \varphi, \quad B(\gamma_2) = U.$$

*We have $\tilde{U} = (A, \Delta)\tilde{\cup}(B, \Delta)$ whereas $(A, \Delta)$, $(B, \Delta)$ are disjoint SS-sd-sets. Hence, $\tilde{U}$ is an SS-sd-disconnected. On the other side, the parametric supra topological spaces related to $(U, \mu, \Delta)$, where: $(U, \mu_{\gamma_1}) = \{U, \varphi, \{f_1, f_2\}, \{f_2, f_3\}, \{f_1, f_3\}\}$ and $(U, \mu_{\gamma_2}) = \{U, \varphi, \{f_1, f_3\}, \{f_2, f_3\}\}$ are SS-sd-connected.*

**(2)** *Let $U = \{f_1, f_2\}$, $\Delta = \{\gamma_1, \gamma_2\}$ and $\mu = \{\tilde{U}, \tilde{\varphi}, (Z_i, \Delta), i = 1, 2, .., 6\}$ be an SSTS on U, where:*

$$Z_1(\gamma_1) = \{f_1\}, \quad Z_1(\gamma_2) = \varphi.$$

$$Z_2(\gamma_1) = \{f_1\}, \quad Z_2(\gamma_2) = \{f_1\}.$$

$$Z_3(\gamma_1) = \{f_1\}, \quad Z_3(\gamma_2) = U.$$

$$Z_4(\gamma_1) = U, \quad Z_4(\gamma_2) = \varphi.$$

$$Z_5(\gamma_1) = U, \quad Z_5(\gamma_2) = \{f_2\}.$$

$$Z_6(\gamma_1) = U, \quad Z_6(\gamma_2) = \{f_1\}.$$

*Hence, $\tilde{U}$ is an SS-sd-connected. On the other side, we have $(U, \mu_{\gamma_2}) = \{U, \varphi, \{f_1\}, \{f_2\}\}$ whereas $U = \{f_1\} \cup \{f_2\}$ in which $\{f_1\}$, $\{f_2\}$ are disjoint supra sd-sets. Therefore, $(U, \mu_{\gamma_2})$ is an SS-sd-disconnected.*

**Definition 4.22** [60] *The set of all SS-sd-boundary points of a soft subset $(T, \Delta)$ of an SSTS $(U, \mu, \Delta)$, denoted by $b_{sd}^s(T, \Delta)$, given by $b_{sd}^s(T, \Delta) = cl_{sd}^s(T, \Delta) - int_{sd}^s(T, \Delta)$.*

**Theorem 4.23** *An SSTS $(N, \mu, \Delta)$ is SS-sd-connected iff every non-null proper subset $(G, \Delta)$ of $\tilde{N}$ has a non-null SS-sd-boundary.*

**Proof. Necessity**: Assume conversely $b_{sd}^s(G, \Delta) = \tilde{\varphi}$, then $cl_{sd}^s(G, \Delta) = (G, \Delta) = int_{sd}^s(G, \Delta)$. Hence, $(G, \Delta)$ is a proper soft subset of $\tilde{N}$ which is both SS-sc-set and SS-sd-set, which contradicts that $\tilde{N}$ is an SS-sd-connected.

**Sufficient**: Suppose conversely $\tilde{N}$ is an SS-sd-disconnected, then there is a proper soft subset $(G, \Delta)$ of $\tilde{N}$ which is both SS-sc-set and SS-sd-set, from Theorem 4.15 (4). It follows that, $cl_{sd}^s(G, \Delta) = (G, \Delta) = int_{sd}^s(G, \Delta)$, and so $b_{sd}^s(G, \Delta) = \tilde{\varphi}$.

**Theorem 4.24** Let $(E, \Delta)$ and $(R, \Delta)$ be two called SS-sd-disconnection of an SSTS $(U, \mu, \Delta)$. For every SS-sd-connected subset $(X, \Delta)$ of $\tilde{U}$, either $(X, \Delta) \tilde{\subseteq} (E, \Delta)$ or $(X, \Delta) \tilde{\subseteq} (R, \Delta)$.

**Proof**. Assume that $(E, \Delta)$ and $(R, \Delta)$ be an SS-sd-disconnection of an SSTS $(U, \mu, \Delta)$, then $\tilde{U} = (E, \Delta \tilde{\cup} (R, \Delta), (E, \Delta)$ and $(R, \Delta)$ are two non-null SS-sd-separated sets. Since $(X, \Delta)$ is an SS-sd-connected subspace of $\tilde{U}$,

$$(X, \Delta) = [(X, \Delta) \tilde{\cap} (E, \Delta)] \tilde{\cup} [(X, \Delta) \tilde{\cap} (R, \Delta)].$$

Now,

$$[(X, \Delta) \tilde{\cap} (E, \Delta)] \tilde{\cup} cl_{sd}^s[(X, \Delta) \tilde{\cap} (R, \Delta)] \tilde{\subseteq} (E, \Delta) \tilde{\cap} cl_{sd}^s(R, \Delta) = \tilde{\varphi} \text{ and}$$

$$cl_{sd}^s[(X, \Delta) \tilde{\cap} (E, \Delta)] \tilde{\cup} [(X, \Delta) \tilde{\cap} (R, \Delta)] \tilde{\subseteq} cl_{sd}^s(E, \Delta) \tilde{\cap} (R, \Delta) = \tilde{\varphi}.$$

This means that,

$$(X, \Delta) \tilde{\cap} (E, \Delta) \text{ and } (X, \Delta) \tilde{\cap} (R, \Delta) \text{ are SS-sd-disconnection for } (X, \Delta),$$

which is a contradiction with the SS-sd-connectedness of $(X, \Delta)$. Therefore, either $(X, \Delta) \tilde{\cap} (E, \Delta) = \tilde{\varphi}$ or $(X, \Delta) \tilde{\cap} (R, \Delta) = \tilde{\varphi}$, from Corollary 4.19. Thus, either $(X, \Delta) \tilde{\subseteq} (E, \Delta)$ or $(X, \Delta) \tilde{\subseteq} (R, \Delta)$.

**Corollary 4.25** *If a soft subsets $(G, \Delta)$ of an SSTS $(U, \mu, \Delta)$ is SS-sd-connected, then $cl_{sd}^s(G, \Delta)$ is also.*

**Proof**. Assume conversely that $cl_{sd}^s(G, \Delta)$ is an SS-sd-disconnected, then there is an SS-sd-disconnection $(E, \Delta)$ and $(R, \Delta)$ for $cl_{sd}^s(G, \Delta)$. It follow that,

$$cl_{sd}^s(G, \Delta) = (E, \Delta) \tilde{\cup} (R, \Delta).$$

Since $(G, \Delta)$ is an SS-sd-connected,

$$\text{either } (G, \Delta) \tilde{\subseteq} (E, \Delta) \text{ or } (G, \Delta) \tilde{\subseteq} (R, \Delta) \text{ from Theorem 4.24.}$$

If $(G, \Delta) \tilde{\subseteq} (E, \Delta)$, then $cl_{sd}^s(G, \Delta) \tilde{\subseteq} cl_{sd}^s(E, \Delta)$ which follows

$$cl_{sd}^s(G, \Delta) \tilde{\cap} (R, \Delta) = \tilde{\varphi}.$$

However, $(R, \Delta) \tilde{\subseteq} cl_{sd}^s(G, \Delta))$. Therefore, $(R, \Delta) = \tilde{\varphi}$, which is a contradiction. By a similar argument, if $(G, \Delta) \tilde{\subseteq} (R, \Delta)$, we can obtain $(E, \Delta) = \tilde{\varphi}$, which is also a contradiction. Consequently, $cl_{sd}^s(G, \Delta))$ is an SS-sd-connected.

**Proposition 4.26** *The soft union of any family of SS-sd-connected subsets $(K_i, \Delta)$, $i \in I$ of an SSTS $(U, \mu, \Delta)$ in which having a non-null soft intersection is an SS-sd-connected set.*

**Proof**. Assume conversely that $(K, \Delta) = \tilde{\bigcup}_{i \in I}(K_i, \Delta)$ is an SS-sd-disconnected, then $(K, \Delta) = (C, \Delta) \tilde{\cup} (D, \Delta)$, where $(C, \Delta), (D, \Delta)$ are SS-sd-separated subsets of $\tilde{U}$. Since $\tilde{\bigcap}_{i \in I}(K_i, \Delta) \neq \tilde{\varphi}$, there exists $s_{\gamma_1} \tilde{\in} \tilde{\bigcap}_{i \in I}(K_i, \Delta)$. It follows that, $s_{\gamma_1} \tilde{\in} (K, \Delta)$. So,

either $s_{\gamma_1} \tilde{\in} (C, \Delta)$ or $s_{\gamma_1} \tilde{\in} (D, \Delta)$.

If $s_{\gamma_1} \tilde{\in} (C, \Delta)$, then

$$s_{\gamma_1} \tilde{\in} (K_i, \Delta) \ \forall i \in I \text{ and } (K_i, \Delta) \tilde{\subseteq} (K, \Delta).$$

Therefore, $(K_i, \Delta) \forall i \in I$ is an SS-sd-connected subset of an SS-sd-disconnected space. Hence,

$$\text{either } (K_i, \Delta) \tilde{\subseteq} (C, \Delta) \text{ or } (K_i, \Delta) \tilde{\subseteq} (D, \Delta) \ \forall i \in I, \text{ from from Theorem 4.24.}$$

If $(K_i, \Delta) \tilde{\subseteq} (C, \Delta) \ \forall i \in I$, then $(K, \Delta) \tilde{\subseteq} (C, \Delta)$ which owing to $(D, \Delta) = \tilde{\varphi}$, which is a contradiction. By a similar argument, if $(K_i, \Delta) \tilde{\subseteq} (D, \Delta) \ \forall i \in I$, we can obtain $(C, \Delta) = \tilde{\varphi}$, which is also a contradiction. Thus, $(K, \Delta)$ is an SS-sd-connected.

**Theorem 4.27** *Let* $(A, \Delta), (B, \Delta)$ *be soft subsets of an SSTS* $(U, \mu, \Delta)$ *such that*

$$(B, \Delta) \tilde{\subseteq} (A, \Delta) \tilde{\subseteq} cl^s_{sd}(B, \Delta).$$

*If* $(B, \Delta)$ *is an SS-sd-connected, then* $(A, \Delta)$ *is also.*

**Proof**. Suppose conversely $(A, \Delta)$ is an SS-sd-disconnected, then there exist two non-null SS-sd-subsets of $(E, \Delta)$ and $(R, \Delta)$ such that $(A, \Delta) = (E, \Delta) \tilde{\cup} (R, \Delta)$. Since $(B, \Delta)$ is an SS-sd-connected and $(B, \Delta) \tilde{\subseteq} (A, \Delta)$, either $(B, \Delta) \tilde{\subseteq} (E, \Delta)$ or $(B, \Delta) \tilde{\subseteq} (R, \Delta)$, from Theorem 4.24. If $(B, \Delta) \tilde{\subseteq} (E, \Delta)$, then $cl^s_{sd}(B, \Delta) \tilde{\subseteq} cl^s_{sd}(E, \Delta)$ which follows

$$cl^s_{sd}(B, \Delta) \tilde{\cap} (R, \Delta) = \tilde{\varphi}.$$

However,

$$(R, \Delta) \tilde{\subseteq} (A, \Delta) \tilde{\subseteq} cl^s_{sd}(B, \Delta).$$

Hence, $(R, \Delta) = \tilde{\varphi}$, which is a contradiction. Thus, $(A, \Delta)$ is an SS-sd-connected. By a similar way, if $(B, \Delta) \tilde{\subseteq} (R, \Delta)$, then $(E, \Delta) = \tilde{\varphi}$ which is also a contradiction. Therefore, $(A, \Delta)$ is an SS-sd-connected.

**Theorem 4.28** *If for all pair of distinct soft points* $s_{\gamma_1}$, $t_{\gamma_2}$ *in an SSTS* $(U, \mu, \Delta)$ *there exists an SS-sd-connected subset* $(H, \Delta)$ *of* $\tilde{U}$ *with* $s_{\gamma_1}$, $t_{\gamma_2} \tilde{\in} (H, \Delta)$, *then* $\tilde{U}$ *is an SS-sd-connected.*

**Proof**. Assume conversely that $\tilde{U}$ is an SS-sd-disconnected, then $\tilde{U} = (M, \Delta) \tilde{\cup} (N, \Delta)$, for some $(M, \Delta), (N, \Delta)$ SS-sd-separated sets. Since $(M, \Delta) \tilde{\cap} (N, \Delta) = \tilde{\varphi}$, there exist a pair of distinct soft points $s_{\gamma_1}$, $t_{\gamma_2}$ such that $s_{\gamma_1} \tilde{\in} (M, \Delta)$ and $t_{\gamma_2} \tilde{\in} (N, \Delta)$. It follows that,

$$s_{\gamma_1}, \ t_{\gamma_2} \tilde{\in} \tilde{U} \text{ with } s_{\gamma_1} \neq t_{\gamma_2}.$$

From hypothesis, there exists an SS-sd-connected set $(H, \Delta) \tilde{\subseteq} \tilde{U}$ with $s_{\gamma_1}$, $t_{\gamma_2} \tilde{\in} (H, \Delta)$.

Hence, either $(H, \Delta) \tilde{\subseteq} (M, \Delta)$ or $(H, \Delta) \tilde{\subseteq} (N, \Delta)$, from Theorem 4.24. If $(H, \Delta) \tilde{\subseteq} (M, \Delta)$, then $s_{\gamma_1}$, $t_{\gamma_2} \tilde{\in} (M, \Delta)$ and so $(M, \Delta) \tilde{\cap} (N, \Delta) \neq \tilde{\varphi}$, which contradicts our hypothesis. Also, we will get the same contradiction if $(H, \Delta) \tilde{\subseteq} (N, \Delta)$. Therefore, $\tilde{U}$ is an SS-sd-connected.

**Theorem 4.29** *The image of an SS-sd-connected set under an SS-sd-irresolute map is an SS-sd-connected.*

**Proof**. Obvious from Proposition 4.12.

**Corollary 4.30** *If $\psi_{sd}$: $(U, \tau, \Delta) \to (V, \sigma, \Lambda)$ is surjective SS-sd-irresolute map with $\tau \subset \mu$ and $\sigma \subset \mu^*$, and $\tilde{U}$ is an SS-sd-connected, then also $\tilde{V}$.*

**Proof**. Obvious from Theorem 4.29.

## 5 Conclusion and upcoming research

In this paper we introduce new approaches of soft continuity in the frame of SSTSs. We provid many characterizations of these notions with the support of SS-closure operator and SS-interior operator. Also, we investigate many properties to the SS-open (closed) maps. Furthermore, we introduce the concept of connectedness inspired by the SS-sd-separated sets. We prove that, this definition is a generalization to the previous such notion. Moreover, we show that, there is no priori relationship between SS-sd-connected SSTS and its parametric supra topological spaces in general. Finally, we prove that the image of an SS-sd-connected set under an SS-sd-irresolute map is an SS-sd-connected. With the support of many concrete examples and counterexamples the outline of the manuscript was considered. Our upcoming project is to introduce and investigate more topological properties such as compactness and paracompactness based on the above-mentioned approaches. Moreover, we will generalize theses notion to the fuzzy supra soft topological spaces [61].

## Acknowledgments

The authors extend their appreciation to the Deanship of Scientific Research at Northern Border University, Arar, KSA for funding this research work through the project number "NBU-FFR-2024-2727-03". Also, this study is supported via funding from Prince Sattam bin Abdulaziz University project number (PSAU/2024/R/1445) and this research is funded partially by Zarqa University Jordan.

## Author Contributions

**Conceptualization:** Alaa M. Abd El-latif, Radwan Abu-Gdairi, M. Aldawood.

**Data curation:** Alaa M. Abd El-latif.

**Formal analysis:** Alaa M. Abd El-latif.

**Funding acquisition:** Alaa M. Abd El-latif, Radwan Abu-Gdairi.

**Investigation:** Alaa M. Abd El-latif, Radwan Abu-Gdairi, Mesfer H. Alqahtani.

**Methodology:** Alaa M. Abd El-latif, A. A. Azzam, Mesfer H. Alqahtani.

**Resources:** Mesfer H. Alqahtani.

**Supervision:** Alaa M. Abd El-latif, A. A. Azzam.

**Validation:** Alaa M. Abd El-latif, M. Aldawood, Mesfer H. Alqahtani.

**Visualization:** Alaa M. Abd El-latif, M. Aldawood.

**Writing – original draft:** Alaa M. Abd El-latif, Radwan Abu-Gdairi, M. Aldawood, Mesfer H. Alqahtani.

**Writing – review & editing:** Alaa M. Abd El-latif, A. A. Azzam, Radwan Abu-Gdairi, Mesfer H. Alqahtani.

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
