## [Decision Letter · Decision Letter 0]

26 Mar 2024

PONE-D-24-09504New versions of maps and connected spaces via supra soft sd-operatorsPLOS ONE

Dear Dr. Azzam,

Thank you for submitting your manuscript to PLOS ONE. After careful consideration, we feel that it has merit but does not fully meet PLOS ONE’s publication criteria as it currently stands. Therefore, we invite you to submit a revised version of the manuscript that addresses the points raised during the review process.

We look forward to receiving your revised manuscript.

Kind regards,

Fucai Lin, Ph.D.

Academic Editor

PLOS ONE

Journal Requirements:

 Whilst you may use any professional scientific editing service of your choice, PLOS has partnered with both American Journal Experts (AJE) and Editage to provide discounted services to PLOS authors. Both organizations have experience helping authors meet PLOS guidelines and can provide language editing, translation, manuscript formatting, and figure formatting to ensure your manuscript meets our submission guidelines. To take advantage of our partnership with AJE, visit the AJE website (http://aje.com/go/plos) for a 15% discount off AJE services. To take advantage of our partnership with Editage, visit the Editage website (www.editage.com) and enter referral code PLOSEDIT for a 15% discount off Editage services. If the PLOS editorial team finds any language issues in text that either AJE or Editage has edited, the service provider will re-edit the text for free.

 A clean copy of the edited manuscript (uploaded as the new *manuscript* file).

“This study is supported via funding from Prince Sattam bin Abdulaziz University project number (PSAU/2024/R/1445) and this research is funded partially by Zarqa UniversityJordan.”

Reviewers' comments:

Reviewer's Responses to Questions

**Comments to the Author**

1. Is the manuscript technically sound, and do the data support the conclusions?

Reviewer #1: Yes

Reviewer #2: Yes

2. Has the statistical analysis been performed appropriately and rigorously? 

Reviewer #1: N/A

Reviewer #2: N/A

3. Have the authors made all data underlying the findings in their manuscript fully available?

Reviewer #1: Yes

Reviewer #2: Yes

4. Is the manuscript presented in an intelligible fashion and written in standard English?

Reviewer #1: Yes

Reviewer #2: Yes

5. Review Comments to the Author

Reviewer #1: This paper belongs to the field of supra soft topology. The authors discussed new types of connectedness and continuity in the spaces of supra soft topologies. They studied their main characterizations with the help of some examples. The obtained results are correct and interesting.

However, the manuscript needs some improvements. In what follows, I mention some revisions that are necessary to improve the manuscript:

1. The quality of the English language should be improved carefully to enhance readability, for example page 2, in the penultimate line of introduction section, write "In addition, we use..." instead of "In addition, we used ...". Also, page 6, write "Let (G, Δ) be both..." instead of "Let (G, Δ) is both ..."

2. Improve the way of presenting literature review, focus on the contributions on supra soft topology the frame of your topic, you should refer to the following (i) Two types of separation axioms on supra soft separation spaces; (ii) On supra soft topological ordered spaces; (iii) Further notions related to new operators and compactness via supra soft topological spaces; (iv) New types of soft ordered mappings via soft $\\alpha$-open sets.

3. Review the proof (2) ⇒ (1) of Theorem 3.7.

4. In Example 4.21, there are two braces for the set (X_4, Δ), correct this typo.

5. I suggest showing the readers how the structures of supra topology are used to address practical problems by referring to the published paper “Rough sets models inspired by supra-topology structures”

6. Improve the way of presentation, write a brief introduction for the main sections.

7. Are all classical properties of the current work still valid in the spaces of supra soft topologies.

8. the conclusion section, mention some future directions and demonstrating the importance of writing the manuscript.

Reviewer #2: After I reviewed this manuscript, I see it needs some revisions and improvements according to the following comments:

1- please, specify the main contributions of this paper.

2- merge paragraphs 3, 4, and 5 into one paragraph.

3- the phrase "(G^c, Δ) is proper supra closed soft subset" is not clear (given in page 6), the correct form is "soft closed" instead of "closed soft"

4-There are some typos in Remark 3.8

5- Link the obtained results with their counterparts in classical settings, demonstrate if there are divergences.

6- It will be good to investigate the importance of studying topological concepts in supra topologies, specify a separate paragraph in the introduction section for this point, show how generalization of topology such supra topology and infra topology will be beneficial for dealing with practical issues by referring to the following manuscripts:

a) Some applications of supra preopen sets

b) Investigation of limit points and separation axioms using supra $\\beta$-open sets

b) Connectedness and covering properties via infra topologies with application to fixed point theorem

c) Continuity and separation axioms via infra topological spaces

6. PLOS authors have the option to publish the peer review history of their article (what does this mean?). If published, this will include your full peer review and any attached files.

Reviewer #1: No

Reviewer #2: No

---

## [Author Response · Author response to Decision Letter 0]

27 Mar 2024

Response to Reviewers

Manuscript ID: PONE-D-24-09504

Title: New versions of maps and connected spaces via supra soft sd-operators

Reply to Reviewer 1:

First of all, we would like to thank you for taking the time and effort necessary to review the manuscript. We sincerely appreciate all valuable comments and suggestions, which helped us to improve the quality of the manuscript.

Secondely, we agree with you. The English language and style were needed to be revised. So, we revised the spelling and grammar errors via professional scientific editing service from Department of Languages and Translation / College of Sciences and Arts / Northern Border University, which is being highlighted in the paper. [In special , See Abstract, Introduction, and Conclusion sections. Also, in the introduction for the main sections].

 Moreover, we improved the manuscript according to your mentioned points as follows:

1. In page 2, we corrected "In addition, we use..." instead of "In addition, we used ..."

2. In page 6, we corrected "Let (G, Δ) be both..." instead of "Let (G, Δ) is both ..."

3. We improved the way of presenting literature review by focusing on the mentioned topics in the Introduction section. [See Introduction section ]

4. We revised the proof (2) ⇒ (1) of Theorem 3.7.

5. We unified the abbreviation for “supra soft” by “SS-” for all the paper.

6. In Example 4.21, the typo of (X4, Δ), is corrected.

7. We investigated the importance of studying topological concepts in supra topologies, in special rough sets models.

8. We wrote a brief introduction for the main sections to improve the way of presentation.

9. [You are right] We linked the obtained results with their counterparts in classical settings, specifically in Remark 4.20 and Examples 4.21.

10. The conclusion section is improved by mention some future directions and demonstrating the importance of writing the manuscript.

11. Depending of the above arguments, the References list is updated.[see the References]

Reply to Reviewer 2:

First of all, we would like to thank you for taking the time and effort necessary to review the manuscript. We sincerely appreciate all valuable comments and suggestions, which helped us to improve the quality of the manuscript.

Secondely, we agree with you. The paper was needed to be revised. So that, we improved the manuscript according to your mentioned notes as follows:

1. We provided the main contributions of this paper in Abstract, Introduction, and Conclusion sections. Also, in the introduction for the main sections].

2. We merged paragraphs 3, 4, and 5 into one paragraph.

3. We unified the abbreviation for “supra soft” by “SS-” for all the paper.

4. In page 5, we replaced “(Gc, Δ) is proper supra closed soft subset” by “(Gc, Δ) is proper SS-closed subset”

5. We corrected the typo in Remark 3.8.

6. [You are right] We linked the obtained results with their counterparts in classical settings, specifically in Remark 4.20 and Examples 4.21.

7. We investigated the importance of studying topological concepts in supra topologies and infra topologies in a separated paragraph in the introduction section and focusing on the mentioned topics. [paragraph 1]

---

## [Decision Letter · Decision Letter 1]

6 May 2024

New versions of maps and connected spaces via supra soft sd-operators

PONE-D-24-09504R1

Dear Dr. Azzam,

We’re pleased to inform you that your manuscript has been judged scientifically suitable for publication and will be formally accepted for publication once it meets all outstanding technical requirements.

Kind regards,

Fucai Lin, Ph.D.

Academic Editor

PLOS ONE

---

## [Editor Report · Acceptance letter]

23 Aug 2024

PONE-D-24-09504R1 

PLOS ONE

Dear Dr. Azzam, 

I'm pleased to inform you that your manuscript has been deemed suitable for publication in PLOS ONE. Congratulations! Your manuscript is now being handed over to our production team.

Kind regards, 

on behalf of

Professor Fucai Lin 

Academic Editor

PLOS ONE